# Type I interferon shapes brain distribution and tropism of tick-borne flavivirus

Nunya Chotiwan [1,2,3,12] ✉, Ebba Rosendal[1,2,12], Stefanie M. A. Willekens [1,2,4,12], Erin Schexnaydre [1,2,5,6,7], Emma Nilsson [1,2], Richard Lindqvist[1,2], Max Hahn [4], Ionut Sebastian Mihai[2,8,9], Federico Morini[4], Jianguo Zhang[1,2,5,6,7], Gregory D. Ebel [10], Lars-Anders Carlson [2,5,6,7], Johan Henriksson [2,7,8], Ulf Ahlgren [4], Daniel Marcellino [11] & Anna K. Överby [1,2] ✉

Viral tropism within the brain and the role(s) of vertebrate immune response to neurotropic flaviviruses infection is largely understudied. We combine multi-modal imaging (cm-nm scale) with single nuclei RNA-sequencing to study Langat virus in wildtype and interferon alpha/beta receptor knockout (*Ifnar*[−/−]) mice to visualize viral pathogenesis and define molecular mechanisms. Whole brain viral infection is imaged by Optical Projection Tomography coregistered to ex vivo MRI. Infection is limited to grey matter of sensory systems in wild-type mice, but extends into white matter, meninges and choroid plexus in *Ifnar*[−/−] mice. Cells in wildtype display strong type I and II IFN responses, likely due to *Ifnb* expressing astrocytes, infiltration of macrophages and *Ifng*-expressing CD8+ NK cells, whereas in *Ifnar*[−/−], the absence of this response contributes to a shift in cellular tropism towards non-activated resident microglia. Multimodal imaging-transcriptomics exemplifies a powerful way to characterize mechanisms of viral pathogenesis and tropism.

Viruses from multiple families are able to cross the physical brain barriers, infect the central nervous system (CNS) and cause disease[1,2]. These include several neurotropic viruses from the genus *Flavivirus*, such as West Nile virus (WNV), Zika virus (ZIKV), Japanese encephalitis virus (JEV) and tick-borne encephalitis virus (TBEV). These flaviviruses frequently cause severe diseases in humans, including encephalitis and meningitis, and lead to significant mortality and long-term neurological sequelae among survivors[3–5]. While global disease burden remains high, no direct treatment for these viral infections is currently available. Consequently, patients rely solely on symptomatic treatment to moderate clinical disease and innate and acquired immune responses

to respond to infection, ultimately resulting in virus clearance. Therefore, a better understanding of viral pathogenesis and the triggered immune response is required for the development of additional diagnostic tools, effective therapies, and preventive measures[6,7]

Induction of type I interferon (IFN-I) is a critical part of the host innate immune response, an important first-line defense against viral infection[8]. The IFN-I response can be activated locally in the CNS and is essential to protect the brain against neurotropic flaviviruses[9–11]. In vitro studies have demonstrated that the IFN-I response is rapidly induced in astrocytes upon TBEV infection and this response protects neurons from infection[12,13]. In models with a deficient or compromised

[1]Department of Clinical Microbiology, Umeå University, 90185 Umeå, Sweden. [2]The Laboratory for Molecular Infection Medicine Sweden (MIMS), Umeå University, 90187 Umeå, Sweden. [3]Chakri Naruebodindra Medical Institute, Faculty of Medicine Ramathibodi Hospital, Mahidol University, Samut Prakan 10540, Thailand. [4]Umeå Centre for Molecular Medicine (UCMM), Umeå University, 90187 Umeå, Sweden. [5]Wallenberg Centre for Molecular Medicine, Umeå University, 90187 Umeå, Sweden. [6]Department of Medical Biochemistry and Biophysics, Umeå University, 90187 Umeå, Sweden. [7]Umeå Centre for Microbial Research, Umeå University, 90187 Umeå, Sweden. [8]Department of Department of Molecular biology, Umeå University, 90187 Umeå, Sweden. [9]Före-tagsforskarskolan, Umeå University, 90187 Umeå, Sweden. [10]Department of Microbiology, Immunology and Pathology, Colorado State University, Fort Collins, CO, USA. [11]Department of Integrative Medical Biology, Umeå University, 90187 Umeå, Sweden. [12]These authors contributed equally: Nunya Chotiwan, Ebba Rosendal, Stefanie M. A. Willekens. ✉e-mail: nunya.chotiwan@umu.se; anna.overby@umu.se

IFN-I response, neurons and astrocytes were more susceptible in vitro[12,14]. In rodent models, this response limits the spread of viral CNS infection and protects mice from lethal encephalitis by Langat virus (LGTV), a less-virulent model for TBEV[9,15]. Furthermore, mice with a deficient IFN-I response showed increased inflammation, blood-brain barrier (BBB) breakdown and increased immune cell infiltration compared to wildtype (WT) mice after LGTV infection[9]. In addition to viral receptors and entry factors, the innate immune response is known to determine cell susceptibility to viral infection[14,16,17], yet how the IFN-I response influences this tropism and shapes global distribution of tick-borne flavivirus infection in the brain is not well understood.

Therefore, we infected WT mice and mice lacking the IFN-I response (*Ifnar*[-/-]) intracranially with LGTV, monitored virus distribution in the brain using advanced imaging techniques and measured cellular responses to infection by single nuclei RNA sequence (snRNAseq). For a holistic view of viral brain infection, we devised an imaging approach that combines viral distribution in the whole brain (~0.5 cm³), obtained by Optical Projection Tomography (OPT)[18], with detailed anatomic information provided by an ex vivo magnetic resonance imaging (MRI)-based brain template. We found sensory grey matter (GM) areas infected in WT mice and this distribution expanded to white matter (WM) in the absence of IFN-I response. Using additional imaging modalities, zooming in to nm scale, we found that the IFN-I response restricts viral replication in meninges and epithelial cells of choroid plexus (ChP). Interestingly, the cellular tropism of LGTV in cerebrum shifted from neurons in WT towards Iba1-positive cells in *Ifnar*[-/-] mice. We then used snRNAseq of cerebral cortex to clarify the transcriptional changes in different cell populations to identify the molecular mechanism of viral pathogenesis. We found that infection in WT mice induced a strong inflammatory response with expression of both type I and type II IFNs (IFN-II), which protected microglia from infection. We also observed infiltration of several distinct immune cell populations, including a large population of peripheral macrophages (MØs). Together, our data provide an unprecedented view of virus distribution and tropism within the brain and highlight the importance of local IFN-I induction in influencing virus distribution, tropism and pathogenesis.

## Results

### Whole brain 3D imaging and coregistration of LGTV infection with MRI in WT and Ifnar[-/-] mice

Local IFN-I restricts neurotropic flavivirus infection in rodent brain as demonstrated by nucleic acid amplification and immunohistochemistry assays[9,12,15,19]. However, these techniques do not provide a comprehensive understanding of viral distribution. Therefore, we set out to establish an imaging approach for neurotropic virus infection in whole mouse brain using OPT, to specifically investigate the effect of IFN-I on viral distribution. To surpass brain barriers, we infected mice intracranially, and brains were collected at humane endpoint, 6–7 days and 4–5 days post-infection for WT and *Ifnar*[-/-], respectively (Supplementary Fig. 1a). Whole brains were bleached to reduce auto-fluorescence, immunolabeled against LGTV nonstructural protein 5 (NS5), mounted in agarose, optically cleared, and scanned using OPT (Fig. 1a, b). Tomographic reconstruction enabled visualization of viral infection throughout the entire brain in 3D (Supplementary Fig. 1b, c). Applying this method, we observed that viral infection in WT brains was predominantly localized in cerebral cortex, with weak signals in the olfactory bulb and with no detectable infection in lateral ventricles nor the fourth ventricle (Fig. 1c inset ii, Supplementary Fig. 1c, and Supplementary Movie 1). In contrast, in *Ifnar*[-/-] brains, we observed weak infection throughout the cerebral cortex but a pronounced infection in the meninges, third ventricle, fourth ventricle, and the interior wall of the lateral ventricles, spanning into the anterior cerebrum and olfactory bulbs, a pattern closely resembling the rostral migratory stream (Fig. 1c inset iii, Supplementary Fig. 1c and Supplementary Movie 1). We also

detected some low-level unspecific signal in uninfected brains (mock group) (Fig. 1c inset i and Supplementary Fig. 1c), indicating possible ventricular antibody trapping or a low, unspecific antibody absorption in meninges during whole brain immunolabeling.

Although OPT enables visualization of LGTV distribution in the whole brain with great sensitivity and high spatial resolution (Fig. 1c), the anatomical information obtained from tissue autofluorescence using this technique is insufficient to provide detailed anatomical information, especially in the cerebrum. As the brain is divided into multiple structures[20], linked to different physiological functions, it would be of great value to improve the anatomical reference frame, to allow precise identification of infected brain areas. To address this, we acquired structural ex vivo MR images from brains after chemical preprocessing and OPT acquisition. These images were coregistered with viral OPT signal, which resulted in fusion images with good spatial alignment (Fig. 1d). In some infected brains, we also observed hyperintense lesions on the T1-weighted images, suggesting virus-induced damages (Fig. 1d inset i). To obtain a reference frame with improved anatomical detail and tissue contrast and to overcome high MRI-scanning costs of individual brains in the future, we used the high-resolution Optically Cleared UMeå (OCUM) brain template, which was generated from MR images acquired after clearing for optical imaging[21]. In optimized OPT-OCUM fusion images, viral signal is displayed within a detailed anatomical context, in 3D, where neuronal pathways and trajectories can be observed, *e.g.*, the rostral migratory stream in an *Ifnar*[-/-] infected brain (Fig. 1e).

### IFN-I response restricts viral infection in meninges and ChP

In WT brains, we observed weak viral signal in the meninges and the interior wall of third ventricle on OPT (Fig. 2 and Supplementary Fig. 1c). However, we did not detect viral antigen in these regions by confocal microscopy (Fig. 2b, d), indicating non-specific antibody trapping in WT mice. In *Ifnar*[-/-] brains, on the other hand, we observed intense viral OPT signal within meninges, which was confirmed by confocal microscopy (Fig. 2b). Additionally, we observed strong viral signal located in all ventricles (Fig. 1c inset iii, 2a). To complement OPT data, we used light sheet fluorescence microscopy (LSFM) to obtain higher resolution images of viral signal in the fourth ventricle and found that ChP, the secretory tissue that resides within ventricles, was highly infected (Fig. 2c and Supplementary Movie 2). Confocal analysis confirmed that ChP of all ventricles, as well as the ependymal cells lining the ventricular walls, were highly infected in *Ifnar*[-/-] brains, but not in WT (Fig. 2d). To further evaluate cellular tropism within ChP, we stained *Ifnar*[-/-] brains with different cellular markers and the viral antigen (NS5) and found that viral infection colocalized with the epithelial cell marker aquaporin-1 (AQP1) in ChP (Fig. 2e). This finding is different from what was recently shown in ZIKV infected mouse brains, where ZIKV specifically targeted pericytes[22].

To further investigate the subcellular localization of virus particles within ChP epithelial cells, we imaged fourth ventricle ChP of *Ifnar*[-/-] brain using electron microscopy. Transmission electron microscopy (TEM) of ChP epithelial cells, identified by the presence of cilia, revealed an extreme distortion of endoplasmic reticulum (ER) membranes in infected tissue (Fig. 2f), in line with previous findings for other flaviviruses in cell culture[23–26]. Higher resolution imaging revealed virus particles in the ER lumen (Fig. 2g, inset i) and formation of viral replication complexes as bud-like invaginations of the ER membrane (Fig. 2g, inset ii). Segmentation of volumetric images, taken by focused ion beam milling-scanning electron microscopy (FIB-SEM), depict the 3D architecture of replication complexes (Fig. 2h-i). Replication vesicles were clustered together within a dilated ER, and virions were detected both within the infected cell and between two cells (visualized by green and blue; Fig. 2j), indicating active viral replication and viral spread within the tissue. TEM and FIB-SEM support earlier observations, confirming productive infection of epithelial cells of ChP

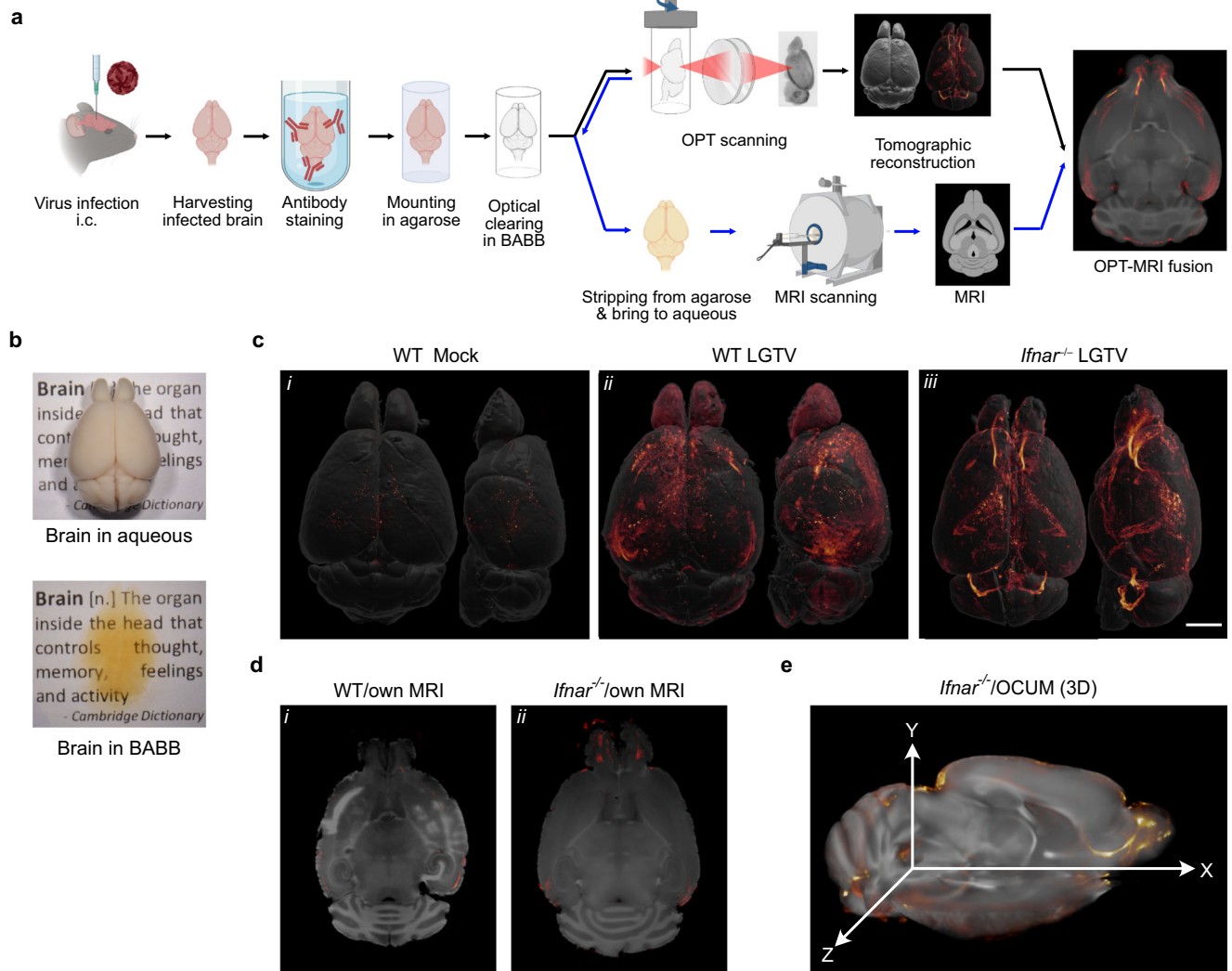

**Fig. 1 | Whole brain 3D imaging of viral infection. a** A workflow for viral infection and whole brain imaging. Illustration was created using BioRender.com. **b** Mouse brain appearance in aqueous and in optical clearing solvent, BABB. **c** Volumetric 3D-render of the brain of mock and LGTV infected brains immunolabeled with anti-NS5 antibody (virus marker; red glow). The signal intensity was normalized within an individual brain and adjusted to identical minimum and maximum. For each image pair, the top and lateral views of the same specimen are shown. Five mice per group were analyzed. Representative images are shown here, and the remaining brains (*n* = 4) are shown in Supplementary Fig. 1b. Scale bar = 2000 μm. **d** OPT-MRI fusion image created using viral OPT signal (red) with own MR scan of WT and *Ifnar*[-/-] brains. **e** 3D mapping of viral OPT signal with the OCUM template.

in *Ifnar*[-/-] brains and provide insight into cytoarchitectural changes induced by viral infection in vivo. Taken together, the observed differences between WT and *Ifnar*[-/-] mice indicate that the IFN-I response efficiently restricts viral replication in the meninges, ventricles and ChP.

## Neuronal circuit mapping and shift in cellular tropism of viral infection

Whole brain OPT-MRI enabled us to visualize, map and quantify viral brain distribution in distinct areas (Fig. 3). Using the OCUM atlas, with 336 distinct anatomical regions[21] (Fig. 3a), we first analyzed the viral signal in the fourth ventricle (Fig. 3b, c, Supplementary Data 1) as proof of principle and saw that the viral signal in *Ifnar*[-/-] is higher compared to WT (*p* = 0.002). Next, we analyzed the viral signal in cortex and olfactory bulb. In WT mice, viral signal was visually detected in entorhinal cortex, piriform area, and primary auditory cortex, which are parts of the olfactory and auditory systems. These regions reside in GM (Fig. 3d, left panel), which consists of neuronal cell bodies. In contrast, *Ifnar*[-/-] mice displayed a more widespread distribution of viral infection, including GM and WM (Fig. 3d). In GM, we observed

viral signal within the olfactory bulb, entorhinal cortex, dorsal endo-piriform nucleus and piriform area. In WM, we detected viral signal within the olfactory limb of the anterior commissure, the lateral olfactory tract, anterior forceps of corpus callosum, and supra-callosal WM. These WM infected regions are also part of the olfactory systems. Of note, the lateral ventricle, subependymal layer of olfactory ventricle, and the olfactory limb of the anterior commissure are all parts of the rostral migratory stream, which suggests specific LGTV infection of the ventriculo–olfactory neurogenic system in *Ifnar*[-/-] brains. It should be noted, however, that large individual differences in viral OPT signal were observed in both WT and *Ifnar*[-/-] mice (Supplementary Fig. 1c, Supplementary Data 1), which, in combination with the relatively small sample size, was also reflected in the viral OPT quantification. In that sense, not all brain regions displaying visual viral signal survived thresholding in quantification (Supplementary Data 1). Nevertheless, OPT quantification confirmed the differences in viral infection between WT and *Ifnar*[-/-] mice (Fig. 3e–l, Supplementary Data 1). In the cortex, WT mice were highly infected in the primary auditory cortex (*p* = 0.003), ventral secondary auditory cortex (*p* = 0.002), temporal association area (*p* = 0.047) and composite entorhinal cortical region

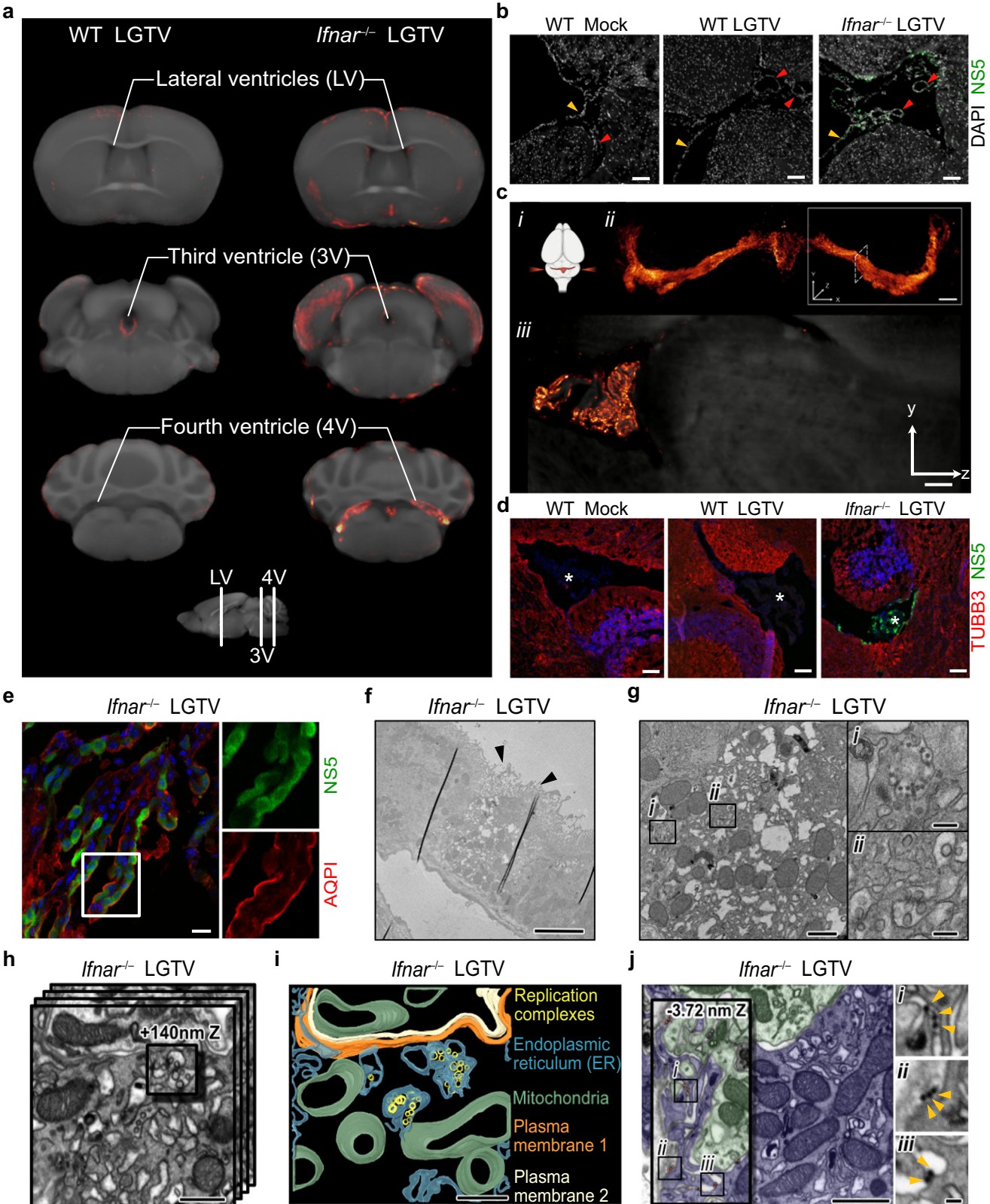

(p = 0.033) (Fig. 3e–g and supplementary Fig. 2a, Supplementary Data 1). However, when taking a closer look at the individual VOIs in the composite entorhinal region, we only observe a significant difference in the medial entorhinal cortex (adjusted p = 0.037) after multiple comparison correction (Supplementary Fig. 2b, Supplementary Data 1). While *Ifnar*−/− mice were significantly more infected in the olfactory peduncle (p = 0.016) (Fig. 3h). *Ifnar*−/− mice displayed more

extensive infection in the olfactory bulb (p = 0.001), subependymal zone of the olfactory ventricle (p = 0.037), and the following WM tracts: olfactory limb of anterior commissure (p = 0.037) and lateral olfactory tract (p = 0.015) (Fig. 3i–l). In the composite hippocampal region, no infection could be detected in WT nor *Ifnar*−/− mice, and only one hippocampal region CA2Or showed infection in WT mice (Supplementary Fig. 2c, d).

**Fig. 2 | IFN-I response restricts viral infection in meninges and ChP. a** OPT-MRI mapping viral infection in the ventricles from OPT signal (red) onto the OCUM template (grey) (*n* = 5). **b** Confocal images of cortical slices showing meninges (yellow arrowheads) and endothelial cells (red arrowheads) immunolabeled with anti-NS5 antibody (green). Scale bars = 100 μm. **c** LSFM showing high-resolution image of the fourth ventricle ChP infected with LGTV, from a representative brain. (i) Tissue orientation during image acquisition. Illustration was created using BioRender.com. (ii) Volumetric 3D render of LSFM of LGTV infected fourth ventricle ChP, immunolabeled with anti-NS5 antibodies (red glow). z = 2000 μm; scale bar = 300 μm. Solid-line square indicates the area imaged and dashed-line shows the orientation of iii. (iii) Tomographic section, YZ plane, z = 600 μm. **d** Confocal images of the fourth ventricle ChP (asterisk). Brains were immunolabeled with anti-NS5 (green), anti-TUBB3 (red) for mature neuron and DAPI for nucleus (blue). Scale bars = 100 μm. **e** Maximum intensity projection capturing fourth ventricle ChP,

immunolabeled anti-NS5 (green) and anti-AQPI (red) for ChP epithelial cell. Scale bars = 20 μm. **f** Ultrastructure of virus replication complex in ChP epithelial cells (indicated by cilia on the apical side; arrowheads) of *Ifnar*⁻/⁻ brain by electron microscopy. Low-magnification TEM image of a heavily infected cell indicated by dilated ER membrane. Scale bar = 5 μm. **g** High-magnification TEM of **f**. Scale bar = 1 μm. Insets show high magnification images of (i) virus particles and (ii) replication complexes. Scale bar = 200 nm. **h–j** FIB-SEM volume imaging shows viral replication complexes within dilated ER. Scale bar = 500 nm. **i** Segmentation image created from the 3D volume images in **h**. Scale bar = 500 nm. **j** Slice-through volume image of two infected cells (blue and green). Scale bar = 1 μm. Inset shows different z depths of the same volume. Boxes *i-iii* show apparent released virus particles (arrowheads) between the cilia. Scale bar = 100 nm. The confocal images *n* = 3/genotype; 2 slices/brain, and the EM images *n* = 2 brains; 3–5 technical replicates. Animations of Fig. 2c and i are shown in the Supplementary Movies 2 and 3.

Cellular tropism in these infected brain regions was further investigated by confocal microscopy. In sections containing areas of rostral migratory stream, we observed viral infection in immature neurons, identified by doublecortin (DCX) immunoreactivity in both WT and *Ifnar*⁻/⁻ brains (Fig. 4a). This indicates that immature neurons are susceptible to viral infection irrespective of type I IFN signaling of the host. On the other hand, no infection was found in astrocytes of either genotype (Fig. 4b). This is in contrast with primary astrocytes in cell culture, which were found to be highly infected by several neurotropic flaviviruses in the absence of IFNAR[12]. Focusing on the cerebrum the total number of infected cells was similar between WT (1631 ± 526) and *Ifnar*⁻/⁻ (1947 ± 213). The majority of infected cells in WT brains were neurons, which have round-cell body morphology and reside in GM, whereas only 3.8 ± 4.8% of all infected cells in cerebrum were positive for the microglial marker Iba1 (Fig. 4c–e). In *Ifnar*⁻/⁻ brains, surprisingly, we observed a shift in cellular tropism from neurons to microglia in which half (50 ± 5.3%) of infected cells were Iba1-positive (Fig. 4c–e). In a highly infected area in *Ifnar*⁻/⁻, the olfactory bulb (Fig. 3i), we also detected prominent microglia infection (Supplementary Fig. 3a), indicating that the tropism shift is not a cortex-specific observation. To further investigate this phenomenon in another system, we isolated primary microglia from neonatal WT and *Ifnar*⁻/⁻ brains and infected them in vitro. Independently of IFNAR protein expression, microglia exhibited low susceptibility to LGTV infection in vitro (Supplementary Fig. 3b).

Taken together, while viral infection is restricted predominantly to neuronal cell bodies of sensory systems of GM in WT, viral tropism expands to WM in the absence of IFN-I signaling. In vivo, cellular tropism was also found to shift towards Iba1-positive cells in the absence of IFN-I signaling, however this tropism shift is not observed in primary microglia from *Ifnar*⁻/⁻ mice in vitro.

### Single nuclei RNA sequencing analysis of virus-induced cellular responses

The increased susceptibility of *Ifnar*⁻/⁻ Iba1-positive microglia in vivo compared to primary microglia in vitro indicates that additional factors of the cellular milieu in the brain influence the tropism of LGTV infection in the absence of IFN-I signaling. Alternatively, this tropism shift might be a result of Iba1 expressing infiltrating MØ that have been detected in the brain after LGTV infection[9]. To investigate these hypotheses, we performed droplet-based single nuclei transcriptomic analysis (10x) of cerebral cortex isolated from LGTV infected WT and *Ifnar*⁻/⁻ mice as well as for uninfected controls (Fig. 5a). After data processing and quality control, we obtained a total of 30,403 nuclei with a median of 1,173 genes detected per nucleus. Using graph-based clustering and analysis of established marker genes, we identified cells of each major cell type expected in cerebral cortex; i.e., neurons, astrocytes, microglia, oligodendrocytes, oligo progenitor cells (OPCs), endothelial cells, vascular leptomeningeal cells (VLMCs) and pericytes (Fig. 5b–d,

Supplementary Data 2). We also detected a small population of *Ttr* expressing cells, most likely representing residual ChP epithelial cells[27]. Interestingly, we also identified cells corresponding to CD8+ NK cells in infected WT brains, but not in *Ifnar*⁻/⁻ or uninfected controls (Fig. 5c). We confirmed the prominent infiltration of CD8+ T-cells, but not CD4+ T-cells or B-cells, in WT mice by qPCR of whole cortex from infected mice (Supplementary Fig. 4a).

Viral RNA in our snRNA data was analyzed by alignment to the TP21 LGTV genome (NC_003690 [https://www.ncbi.nlm.nih.gov/nuccore/NC_003690]). In WT mice, we detected most viral RNA in excitatory neurons and pericytes, while in *Ifnar*⁻/⁻ mice, the LGTV reads were mainly found in Microglia/machrophages (Micro/MØ) and ChP, and to a lesser extent astrocytes (Fig. 5e). This supports our immunofluorescence data (Figs. 2, 4) and indicates that the removal of ER-derived viral replication complexes from nuclei was not complete using the standard nuclear isolation protocol for snRNAseq.

### IFN-I response increases inflammation and immune cell infiltration

To understand the fundamental differences between WT and *Ifnar*⁻/⁻ mice, we next analyzed differential gene expression in all cell types at baseline (uninfected controls). Although we previously reported that primary cell cultures of astrocytes from WT and *Ifnar*⁻/⁻ mice display major gene expression differences at baseline[12], our snRNAseq analysis presented very little differences in vivo (Supplementary Fig. 4b). This suggests that baseline expression of entry factors or restriction factors in microglia is not driving the shift in cellular tropism seen in *Ifnar*⁻/⁻ mice. Subsequently, we evaluated the differentially expressed genes (DEGs) in response to LGTV infection and performed gene set enrichment analysis (GSEA) on the Reactome pathways[28]. In WT brains, we identified a large number of DEGs (2,846), with the majority of expression changes in microglia, astrocytes, oligodendrocytes, VLMC and endothelial cells (Fig. 5f, Supplementary Data 2). Although we detected a similar viral load in cerebral cortex of WT and *Ifnar*⁻/⁻ mice at endpoint (Supplementary Fig. 4c), the response to viral infection was markedly different in WT and *Ifnar*⁻/⁻, and a lower number of DEGs were identified (1,039) in *Ifnar*⁻/⁻ (Fig. 5f, g, Supplementary Data 2). The majority of DEGs for both WT and *Ifnar*⁻/⁻ were found to be upregulated (Supplementary Data 2). We compared the overlap of upregulated DEGs amongst the five most responsive cell types based upon number of DEGs (astro, micro/MØ, oligo, VLMCs and endo) (Fig. 5g). In WT mice, 73 genes, corresponding to 9% of all upregulated DEGs, were commonly upregulated in these cells showing a partial overlap and uniform cell response to infection. Of note, 57% of the genes were not shared, and specific to each cell type and for astro 23%, micro/MØ 48%, oligo 21%, VLMCs 12% and endo 18% of the DEGs were uniquely upregulated, emphasizing the advantage of a snRNAseq approach. In *Ifnar*⁻/⁻, however, only nine genes, corresponding to 3% of all upregulated DEGs were commonly upregulated, indicating that the majority of responses are unique to each cell type.

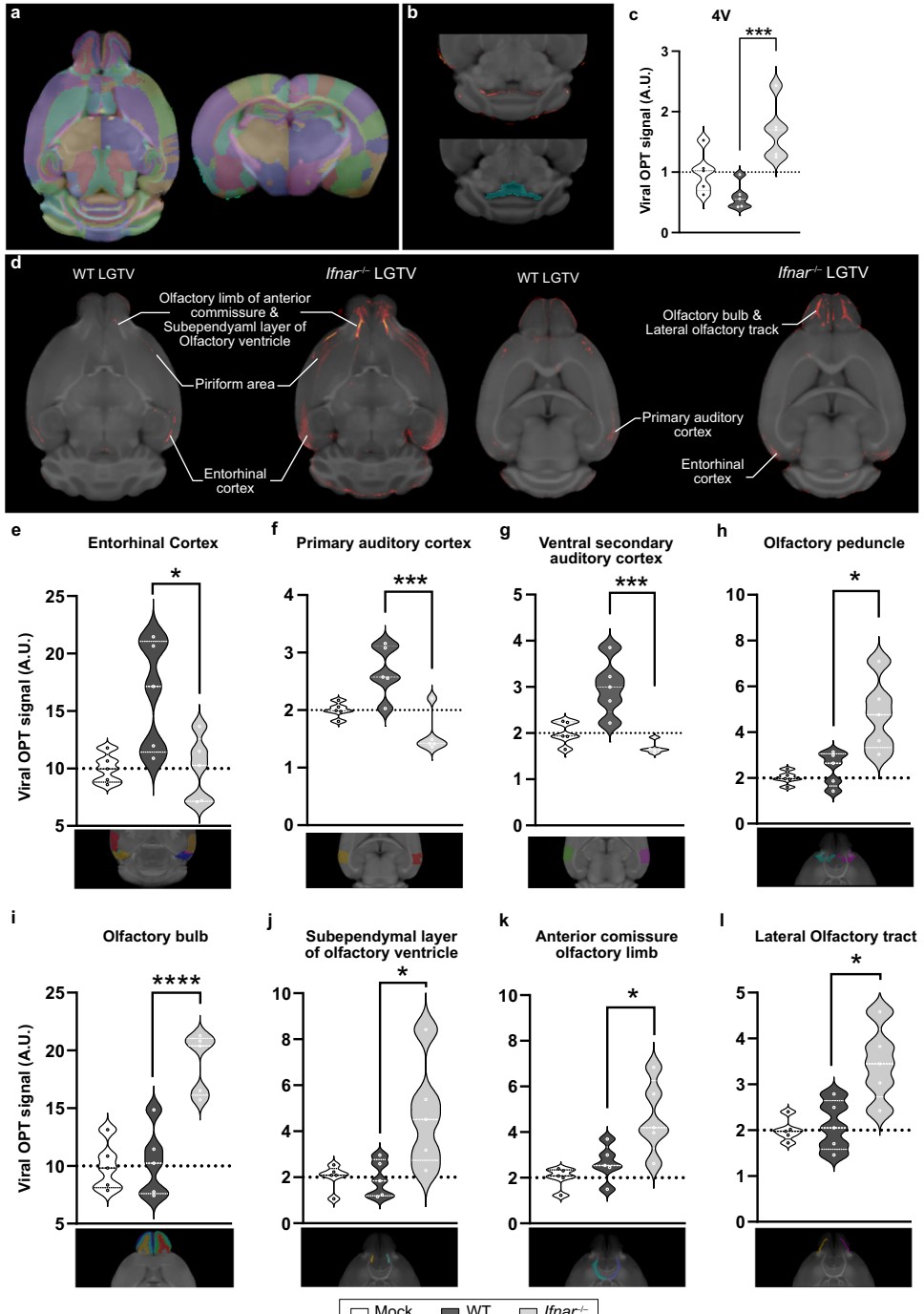

**Fig. 3 | Imaging and quantification of viral infection revealed a shift in infected regions shaped by the IFN-I response. a** Axial and coronal representation of the OCUM brain template with its according atlas[21]. **b** Typical example of an OPT-OCUM fusion displaying signal in the fourth ventricle (upper panel) and the VOI delineation based on the OCUM-atlas (lower panel). **c** Quantification and statistical analysis (two-tailed unpaired t-test) of viral OPT signal in fourth ventricle of WT and *Ifnar*[-/-] mice. All values were normalized to the mock group and are expressed in arbitrary units (A.U.). ***$p = 0.002$. **d** Anatomical mapping of infected cortical and olfactory bulb after coregistration of viral OPT signal (red) with OCUM brain template (grey). Images of WT and *Ifnar*[-/-] infected brains are shown in two representative axial plains. **e–l** Quantification and statistical analysis (two-tailed unpaired t-test) of viral OPT signal in representative VOIs or composite ROIs. For: **e** *$p = 0.033$, **f** ***$p = 0.003$, **g** ***$p = 0.002$, **h** *$p = 0.016$, **i** ****$p = 0.001$, **j** *$p = 0.037$, **k** *$p = 0.037$, **l** *$p = 0.015$. Source data are provided as a Source Data file.

We found a greater number of significant reactome pathways in WT compared to *Ifnar*[-/-] (Fig. 5h, Supplementary Data 2). In WT mice, there was a large overlap between the pathways upregulated in the different cells and, as expected, the IFN signaling pathway was the most upregulated pathway in all cell types (Fig. 5i). It has been reported that some cells are able to mount an IFN-independent upregulation of interferon stimulated genes (ISGs)[29,30]. In line with this, we also detected an IFN-independent upregulation of IFN stimulated genes in ChP, astrocytes and VLMCs (Fig. 5i) in *Ifnar*[-/-] mice. Astrocytes upregulated 18 ISGs previously shown to be IFN independently upregulated[30], and some with known antiviral activity e.g. *Rsad2* (viperin)[14,31], possibly contributing to lower the susceptibility of these cells in vivo (Fig. 4b and Fig. 5e). However, this antiviral IFN-independent response were not detected in IFNAR deficient microglia.

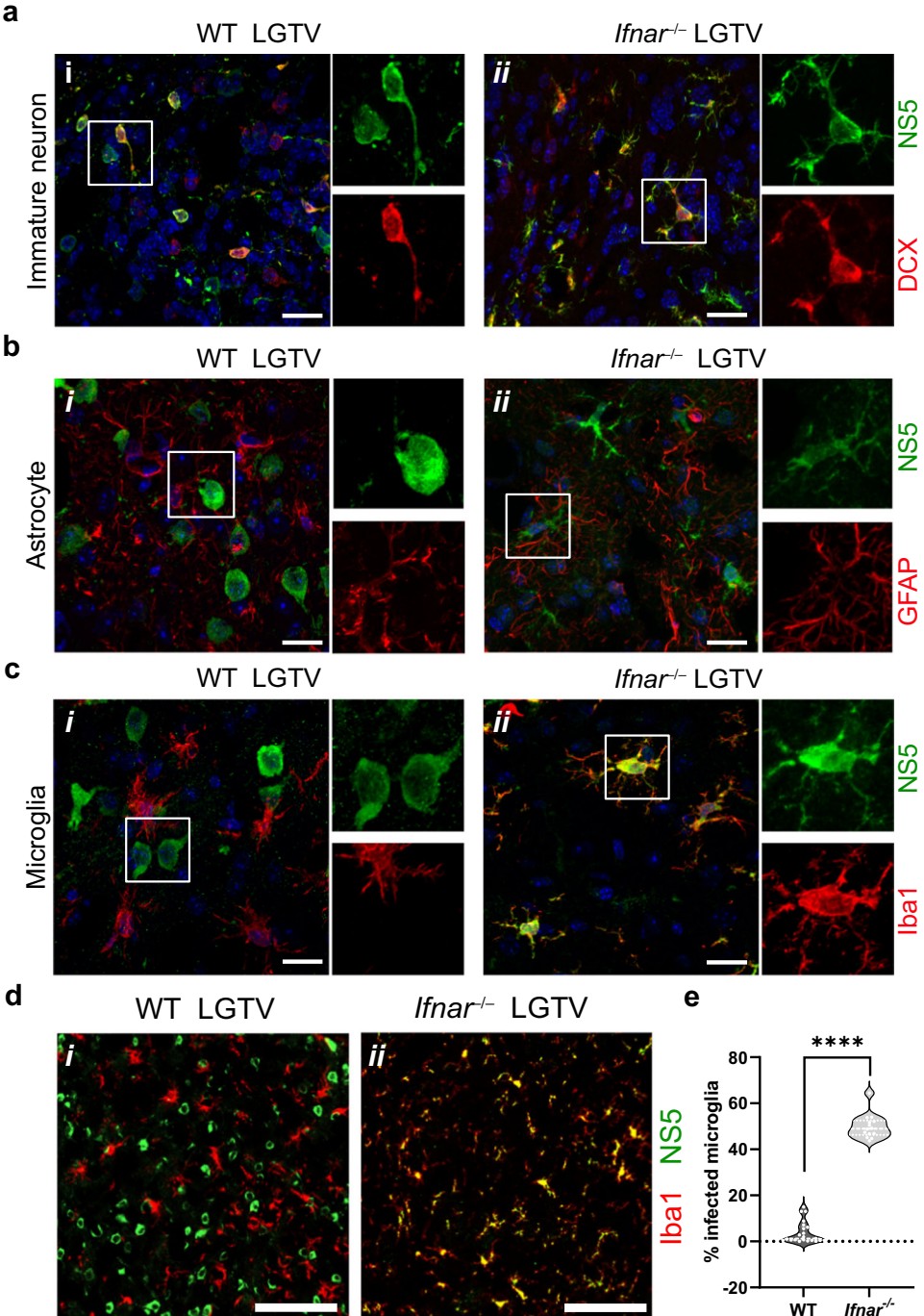

**Fig. 4 | Lack of IFNAR expression shifts cellular tropism from neurons to microglia in the cortex of mice. a–c** Maximum-intensity projection of confocal z-stack. The images were taken from sagittal brain sections (10 μm) using confocal microscope (*n* = 3 per genotype). Scale bars = 20 μm. **a** Captured at the granule cell layer of olfactory bulb. The sections were immunolabeled using anti-NS5 (green), anti-DCX (red) for immature neurons in the rostral migratory stream and DAPI for nucleus (blue). **b** Captured within entorhinal cortex and piriform area. The sections were immunolabeled with anti-NS5 (green), anti-GFAP (red) for astrocyte and DAPI for nucleus (blue). **c** Captured within entorhinal cortex and piriform area. The sections were immunolabeled using anti-NS5 (green), anti-Iba1 (red) for microglia, and DAPI for nucleus (blue). **d** Low magnification images show representative fields of the sagittal slices of cerebral cortex stained with NS5 and Iba1 in WT and *Ifnar*⁻/⁻ brains. Scale bars = 100 μm. **e** Percent of Iba1-positive infected cells of total number of infected cells in cortical slices. Statistical analysis (two-tailed unpaired t-test) showed a significant difference (****$p < 0.0001$) between the percentage infected Iba1-positive cells in *Ifnar*⁻/⁻ mice as compared to WT (*n* = 2–3 per genotype and 6–8 slices per brain). Source data are provided as a Source Data file.

IFNγ is known to compensate for lack of IFN-I, and can exert non-cytolytic antiviral activity against WNV, Sindbis virus and measles virus infection in neurons[32–35]. Moreover, brains infected with Vesicular stomatitis virus have been shown to induce more *Ifng* in the absence of IFNAR signaling compared to WT[29]. Therefore, we specifically evaluated expression changes of IFN-II (*Ifng*)

in whole cortex of mice with qPCR. We found high upregulation of Ifng in WT but not in *Ifnar*⁻/⁻ after infection (Supplementary Fig. 4d). Analyzing the cell types expressing *Ifng* and IFN-Is in WT, we found that particularly CD8+ NK cells strongly expressed *Ifng* while ChP, astrocytes and microglia were the main producers of *Ifnb* (Fig. 5j). One of the main functions of IFN signaling is the

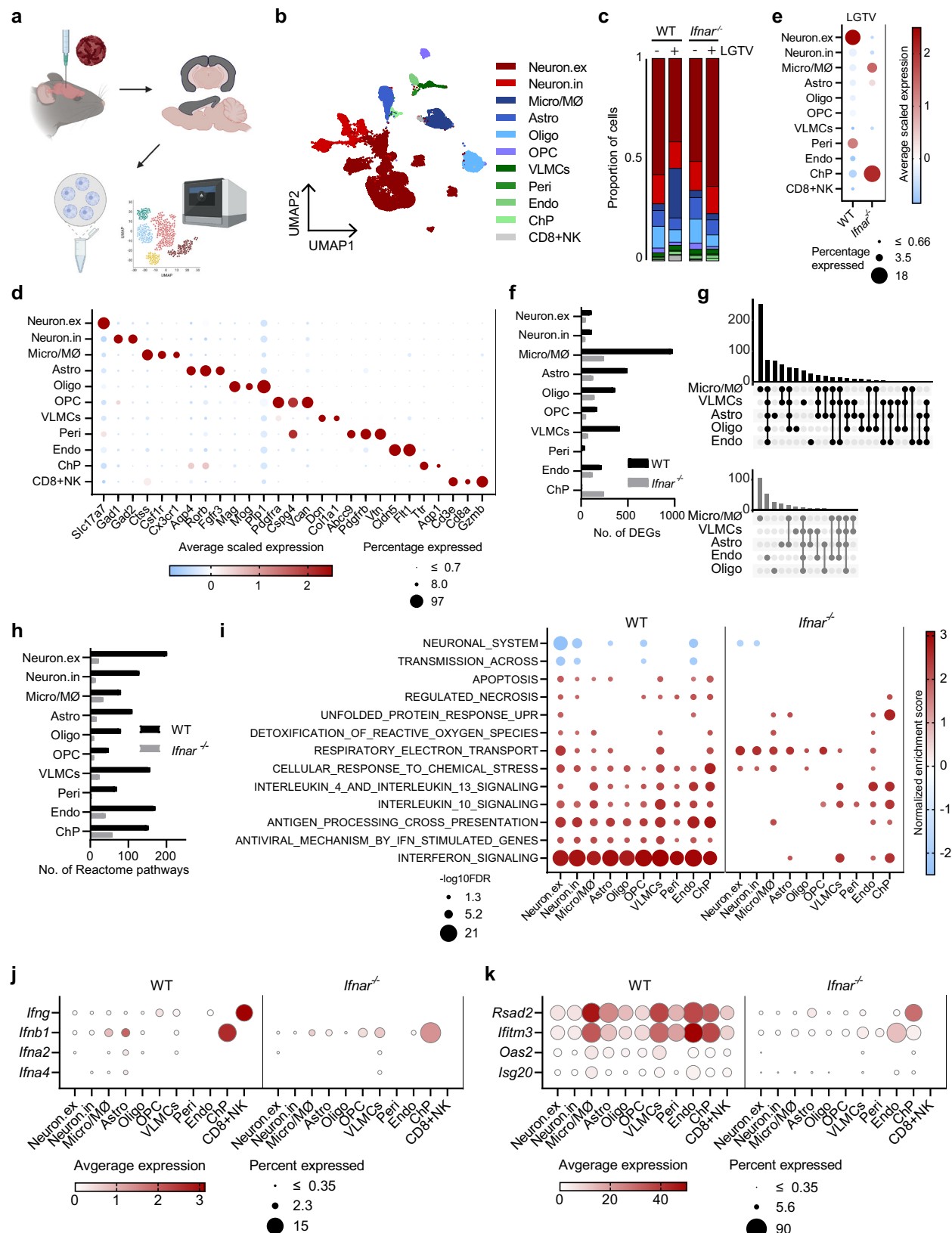

upregulation of antiviral ISGs to limit viral replication and spread. We detected *Rsad2* and *Ifitm3*, two ISGs with confirmed antiviral activity against TBEV[31,36], in all cell types in WT mice, with the strongest expression in microglia, astrocytes, VLMCs, Endo and ChP, whereas only the ChP expressed *Rsad2* and both ChP and Endo expressed *Ifitm3* in *Ifnar*[-/-] mice (Fig. 5k).

Since *Ifng* expressing CD8+ NK cells might contribute to increased inflammation in infected WT mice, we evaluated the expression of known T cell chemoattractants (Supplementary Fig. 4e) and observed an increase in *Ccl2*, *Ccl5* and *Cxcl10* levels in infected WT compared to *Ifnar*[-/-] mice. Furthermore, VLMCs, pericytes and endothelial cells of WT mice expressed high levels of *Cxcl9*, a chemokine dependent on

**Fig. 5 | Single nuclei RNA sequencing (snRNAseq) analysis of LGTV infection shows strong IFN response in the cortex of WT mice. a** Schematic overview of the workflow. For each condition one male and one female mouse were pooled. Illustration was created using BioRender.com. **b** Uniform Manifold Approximation and Projection (UMAP) of 30,403 nuclei captured by droplet based snRNAseq (10x) colored by assigned identities; excitatory neurons; neuron.ex, inhibitory neurons; neuron.in, microglia/macrophages; micro/MØ, astrocytes; astro, oligodendrocytes; oligo, oligodendrocyte progenitor cells; OPC, vascular leptomeningeal cells; VLMCs, pericytes; peri, endothelial cells; endo, natural killer and CD8+ T-cells; CD8 + NK, choroid plexus epithelial cells; ChP. **c** Relative proportion of each cell type in uninfected and LGTV infected samples. **d** Dotplot showing expression of canonical marker genes used to identify major cell types. **e** Dotplot showing LGTV read counts for each cell type. **f** Number of DEGs ($\log_2$FC > 1, $p_{adj}$ < 0.05) induced by LGTV infection in WT or *Ifnar*[-/-] mice. **g** Overlap in upregulated DEGs between five most reactive cell types, shown as a modified UpSet plot for WT and *Ifnar*[-/-] separately. **h** Number of significant Reactome pathways following LGTV infection. **i** DotPlot showing examples of Reactome pathways altered by infection, color corresponding to Normalized Enrichment Score and size corresponding to $-\log_{10}$ adjusted p-value. DotPlot showing expression of IFN-I and IFN-II (**j**) and ISGs (**k**) in infected data sets. Source data are provided as a Source Data file.

IFNγ for cerebral expression[37]. This further confirms the role of both IFN-I and IFN-II signaling in the brain of WT mice. A high expression of corresponding receptors *Ccr2* and *Ccr5* were found in micro/MØ and CD8+ NK cells and *Ccr1* in micro/MØ (Supplementary Fig. 4e). We performed sub-clustering of the infiltrating CD8+ NK cell population but were unable to obtain a clear separation between CD8+ T-cells and NK-cells (Fig. 6a), while flow cytometry confirmed two distinct populations of CD8+ T-cells (CD8$^+$) and NK-cells (CD335$^+$) in infected WT mice (Supplementary Fig. 5a, b). We found granzyme A (*Gzma*), granzyme B (*Gzmb*) and perforin (*Prf1*) to be highly upregulated in these cells in our data set (Supplementary Data 2), which are used by both CD8+ T-cells and NK-cells to kill target cells[38] and may contribute to immunopathology.

### Microglial activation and large infiltration of peripheral macrophages

We detected a major increase of microglia/MØ in WT mice upon infection (from 3.4% to 25%) (Fig. 5c). This suggests that the inflammatory milieu in infected WT mice induces either proliferation of resident microglia and/or infiltration of peripheral MØs. By a sub clustering of the microglia/MØ population, we identified, in addition to microglia (*Cxcr1*) and MØ (*Slfn4*), a cluster of monocyte-like cells (*Ccl3, Ccl4, Il12b* and *Tnf*) and a small number of neutrophils (*S100a9, S100a8* and *Clec4d*) (Fig. 6b, Supplementary Data 2). In *Ifnar*[-/-] mice, we found predominantly microglia (Fig. 6c, left panel), while in WT, we observed a large MØ population during infection (Fig. 6c, right panel). The cellular identity of microglia and MØ was confirmed by expression of cell type specific marker genes[39] (Supplementary Fig. 5c) as well as expression of *Tmem119* (Fig. 6d), a highly specific marker for resident microglia[40]. We used flow cytometry to confirm the presence of infiltrating MØ (CD11b$^+$ CD45$^{high}$) in LGTV infected WT mice, which was not seen in untreated controls or infected *Ifnar*[-/-] mice (Supplementary Fig. 5a, Fig. 6e). Interestingly, although we only detected a small population microglia in infected WT mice in our snRNAseq dataset, the numbers of microglia did not change after infection, as detected with FACS analysis (Fig. 6e). Flow cytometry further confirmed TMEM119 as a microglia specific marker (Fig. 6f) and confocal microscopy showed the presence of infiltrating CD45$^+$ cells in infected WT (Fig. 6g).

We used immunohistochemistry to further investigate the activation and infection status of the microglia/MØ in WT and *Ifnar*[-/-] mice. In WT mice, multiple activation states of uninfected microglia were identified by microglial morphology (Fig. 6h). In *Ifnar*[-/-] mice, the morphology showed non-activated microglia and the majority of LGTV infected cells were positive for TMEM119 (Fig. 6h). However, Iba1-positive TMEM119-negative infected cells were occasionally detected (Fig. 6h). To better understand what drives the differences in microglial activation between WT and *Ifnar*[-/-], we analyzed the cell-cell communication in the snRNA dataset by Cellchat[41]. To look more specifically at microglia, the cells annotated as MØ were separated before analysis. We found fewer overall interactions in infected *Ifnar*[-/-] mice compared to infected WT mice (Fig. 6i), partially explained by the absence of MØ and CD8+ NK cells. Looking at the specific numbers of unique ligand-receptor interactions in WT and *Ifnar*[-/-] going into

microglia cells during infection, the total number of interactions was reduced by 43% in *Ifnar*[-/-] (Fig. 6j and Supplementary Data 2). These interactions are however very different and only 12% of interactions in WT are shared with *Ifnar*[-/-]. Together this suggests that the absence of the local inflammatory milieu (Fig. 5) and the specific cues from other cells that should activate microglia are lost in *Ifnar*[-/-] thus rendering them more susceptible to infection.

## Discussion

While it is clear that the IFN-I response limits viral replication in the brain, viral distribution and cellular tropism on a global whole-brain level have not been addressed, nor their underlying molecular mechanisms. In this study, using multiple imaging modalities (ranging from cm to nm) and single nuclei transcriptomics technology, we evaluated the impact of the IFN-I response on viral distribution within the whole brain with high resolution MRI-guided anatomical precision and the transcriptional changes to identify the molecular mechanism of viral pathogenesis. We show that type I IFN response specifically limits LGTV replication in ChP and cerebral WM. We also show that the specific interplay between cells and the local milieu in vivo, determines the susceptibility to viral infection. In the absence of IFN-I signaling, the complete response to infection is hampered, microglia do not become activated thus rendering them susceptible to infection.

To date, the details regarding viral brain infection and pathogenesis have been partially concealed by an inability to visualize viral infection on the whole organ level. OPT and LSFM are 3D imaging techniques for transparent mesoscopic-sized (mm-cm) tissue samples; and are therefore suitable for rodent whole brain imaging. These methods provide information at cellular resolution while still capturing the entire tissue in 3D and have been used in various applications for targeted imaging (see for example Alanentalo et al. 2007, Hahn et al. 2021 and Dodt et al. 2007[42–44]). However, anatomical information obtained from optical imaging, based on tissue autofluorescence, is limited. MRI, conversely, is widely used for anatomical brain imaging since it has exquisite contrast and resolution[45]. Structural MRI is not suitable for imaging pathogens, but when combined with OPT, one could gain information on both viral distribution and its anatomical location. Several mouse brain templates are available. However, they are either based on histology[20,46] or, in vivo or ex vivo MRI in situ[47–49]. OPT images of whole brain viral signal are acquired ex vivo after tissue bleaching, de-hydration and optical clearing. These chemical treatments result in a certain degree of tissue shrinkage, which leads to misalignment when coregistering brain OPT signals to existing templates. Therefore, we used the OCUM brain template[21], specifically created from ex vivo MR images acquired after OPT pre-processing, yielding improved alignment and precise anatomical mapping of virus distribution in the brain.

Here, we studied the effect of the local IFN-I response on LGTV neurovirulence in the brain after intracranial infection. This model was chosen to eliminate all peripheral IFN effects and to provide equal amounts of virus full access to the brain. OPT-MRI fusion images, together with LSFM, revealed widespread viral infection in olfactory bulb, rostral migratory stream, ventricles, ChP, GM and WM, and showed that whole brain infection patterns are influenced by the IFN-I

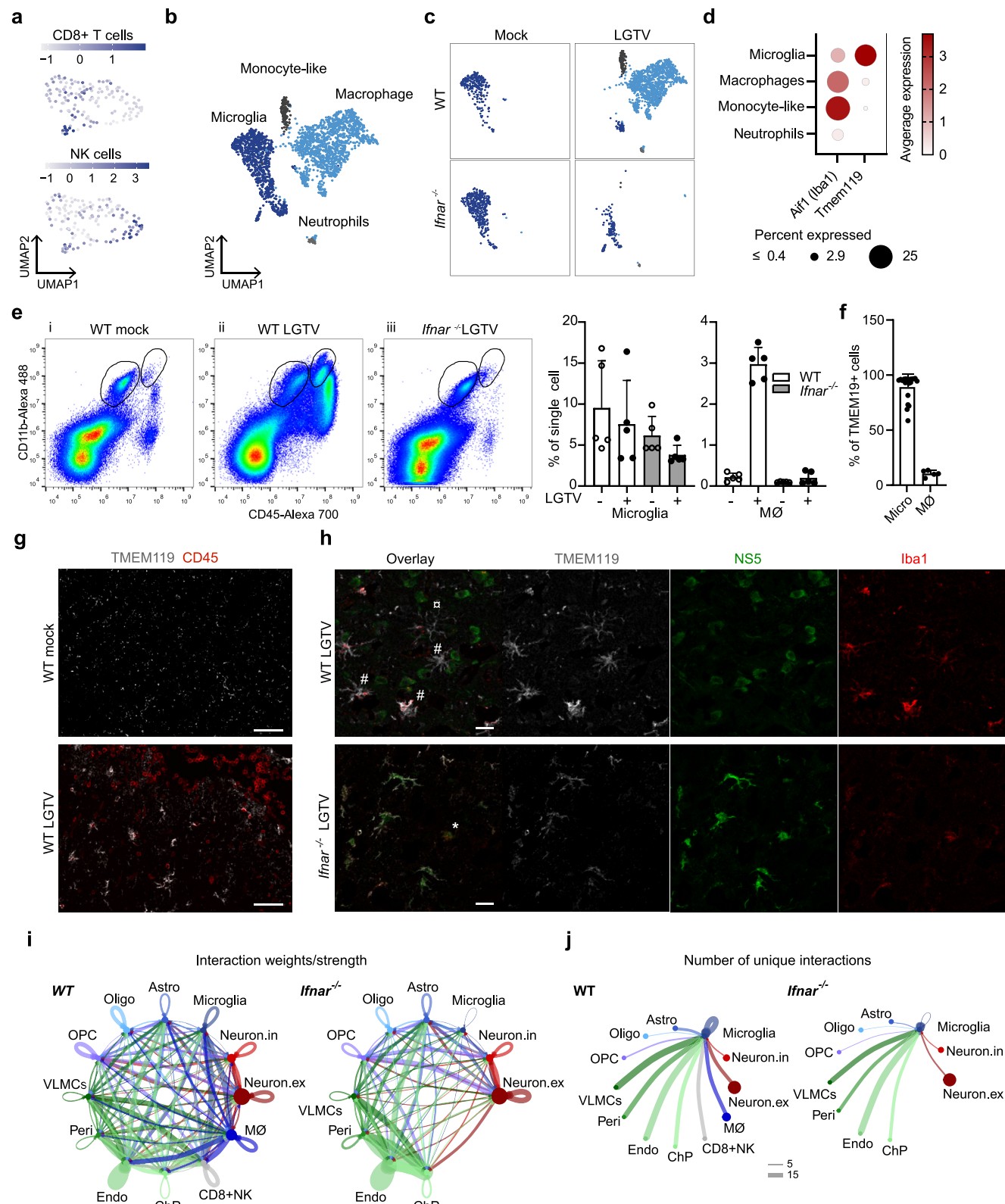

response. Interestingly, at endpoint, meninges and ChP were only infected in *Ifnar*[−/−] brains. This is similar to Herpes Simplex virus I (HSV1) and ZIKV, where viral infection is detected in ChP of *Ifnar*[−/−] brains but not in WT mice[22,50]. This strongly suggests that the IFN-I response plays a major role in preventing viral infection in ChP. Of note, while we detected productive LGTV infection in ChP epithelial cells, a previous study found that ZIKV mainly infects pericytes in ChP[22], indicating tropism differences exist among neurotropic flaviviruses. In cerebrum, infection in both WT and *Ifnar*[−/−] was predominantly restricted to regions processing sensory information, which has been reported previously in animal experiments with other neurotropic flaviviruses, such as TBEV and JEV[51,52]. When comparing *Ifnar*[−/−] and WT visually, we observed that the olfactory system was most infected and widespread in *Ifnar*[−/−] mice, with extensive virus signal in areas processing sensory input from the olfactory bulb such as the granular cell layer of the olfactory bulb. These visual

**Fig. 6 | IFN signaling is important for infiltration of immune cells and activation of microglia after LGTV infection. a** Sub clustering of 144 nuclei of CD8+ NK cell subset found in infected WT, colored by gene signatures for CD8+ T-cells (*Cd3g, Cd3d, Cd28, Cd8a, Cd8b1*) or NK-cells (*Klrblc, Klrd1, Nrc1, Nkg7*). **b** Sub clustering of 2,363 nuclei of micro/MØ cell subset belonging to all datasets, colored by assigned identities visualized by UMAP; Microglia, Macrophages (MØ), Monocyte-like cells and Neutrophils. **c** UMAP from (b) showing nuclei from each condition separately. **d** DotPlot showing expression of *Aif1* (Iba1) and *Tmem119* in cell subtypes. **e** Flow cytometry of brain leukocytes from the cerebral cortex of WT and *Ifnar*$^{-/-}$ mice. Representative scatterplots and percentage microglia (CD11b$^+$, CD45$^{int}$) and MØ (CD11b$^+$, CD45$^{high}$). Data represent the mean ± standard error of the mean (SEM),

$n$ = 5 mice. **f** Percentage of microglia and MØ amongst TMEM119$^+$ cells, in 20 mice (WT and *Ifnar*$^{-/-}$ mock and infected) (microglia) and 5 mice (WT infected) (MØ). Data represent the mean ± SEM Maximum-intensity projection of confocal z-stack captured within the sagittal slices of cortex ($n$ = 3 per group). The sections were immunolabeled anti-CD45 (red) and anti-TMEM119 (white) antibodies (**g**) or anti-NS5 (green), anti-Iba1 (red) and anti-TMEM119 (white) antibodies (**h**) in WT and *Ifnar*$^{-/-}$ brains. ¤ resting state microglia, # different activated sates of resident microglia and * infiltrating macrophage. Scale bar = 60 μm (**g**) or 20 μm (**h**). **i** The communication probability of interactions in infected WT and *Ifnar*$^{-/-}$ data sets and **j** number of unique ligand-receptor interactions going from the different cells to microglia, analyzed by CellChat. Source data are provided as a Source Data file.

observations were also confirmed by the atlas-based quantification of viral signal in the olfactory bulb. Furthermore, VOI-based quantification revealed, in addition to the granular cell layer, clear viral infection of other cell layers as well as in the lateral olfactory tract (Supplementary Data 1), which underlines the advantage of atlas-based OPT analysis over sole-visual observations. While anosmia (loss of smell) is not a common symptom in tick-borne encephalitis patients, nor has it been tested in animal models, a case with anosmia post TBEV vaccination has been reported[53]. Although we could observe viral signal in cortical areas involved in processing and integrating olfactory and gustatory information (piriform cortex, endopiriform nucleus and agranular insula, all located in the orbitofrontal cortex) in some mice. These regions did not survive thresholded quantification, most likely due to the high inter-individual differences in both groups and the relatively small sample size. Nevertheless, endopiriform nucleus consists of specialized cells with long axons projecting towards, among others, the entorhinal cortex, located at the caudal end of the temporal lobe, another cortical area identified to be infected in our study, both visually and quantitatively. Interestingly, OPT quantification showed higher infection in the composite entorhinal region in WT when compared to *Ifnar*$^{-/-}$, which can be mainly attributed to the contribution of the medial entorhinal cortex in the composite entorhinal region (Supplementary Fig. 2b, Supplementary Data 1), and in the primary and secondary auditory cortex of WT mice. Dysfunction of the auditory processing pathway in CNS can lead to sensorineural hearing loss and has been reported, although very rarely, after HSV1 and WNV encephalitis in combination with other severe neurological complications[54–56]. Other flaviviruses such as ZIKV, WNV, and JEV were described to target hippocampal regions, affecting learning and memory[57–59]. We did not see hippocampal infection of LGTV visually, however, hippocampal area CA20r showed infection in WT. It has also been shown by others that LGTV can cause impaired memory function and anxiety–like behavior in WT mice[60].

Interestingly, we observed hyperintense lesions on T1-weighted images of infected brains, which is in line with imaging data obtained from TBE patients, which predominantly show lesions in the basal ganglia, thalami and cerebellum[61]. Interestingly, post-mortem IHC of TBEV brains showed the presence of virus in these regions[62]. However, in our study, the cortical lesions detected on T1 images appeared to be much more widespread than the viral signal detected by OPT. These observations suggest that CNS damage cannot be explained only by the presence of viral infection in a particular area and suggest neuroinflammation-induced lesions.

Single cell techniques are useful in studying the transcriptional response in specific cell types upon infection or inflammation. In contrast to bulk RNA sequencing or microarray analysis which measures the combined response of all cells, single cell techniques better capture cell heterogenicity and genes expressed exclusively by small cell populations. In theory, the single nuclei isolation should only isolate the nuclei from cells. However, we could detect enrichment of LGTV reads in specific populations of cells (excitatory neurons and pericytes in WT and microglia and ChP in *Ifnar*$^{-/-}$). As we show with immunofluorescence that neurons are the main target in WT and

microglia and ChP are infected in *Ifnar*$^{-/-}$, the detected LGTV reads are likely not contamination from free-floating RNA. It is more likely that the ER, which contain viral replication vesicles, is not completely detached from the nuclei during processing. Similar to what has previously been shown for ZIKV[22] and JEV[63,64] we found LGTV reads in pericytes in WT mice. For both ZIKV and JEV the infection of pericytes has been suggested to be important for the virus crossing the blood brain barrier (BBB)[65], as pericytes are crucial for the interaction between astrocytes and the endothelia and thus BBB integrity[66]. In this study we are unable to draw conclusions about how LGTV crosses the brain barriers to enter the CNS as we use intracranial route of infection. However, we do see that depending on the IFN response the virus targets either pericytes in WT or ChP epithelial cells in the *Ifnar*$^{-/-}$.

Several reports have shown IFN-I independent upregulation of ISGs upon infection[29,30,67], In our study this was limited to only a few cell types. In contrast to what we previously showed for astrocytes cultured in vitro[12], we did not detect a difference in baseline gene expression levels between WT and *Ifnar*$^{-/-}$ in vivo. On the same note, we detected a strong tropism shift towards microglia in the cortex of mice, in the absence of IFNAR expression. This was also observed in the olfactory bulb of *Ifnar*$^{-/-}$ mice, where high levels of infection were detected. Nevertheless, IFNAR deficient microglia in monoculture remained resistant to infection in vitro. This might be explained by the inhibitory signals that microglia receive from other cells in vivo[68], and the absence of these in primary monocultures might lead to the "priming" of the microglia to be in a semi-activated state. Others have shown that in order for microglia in the olfactory bulb to become activated they need specific cues from neurons and astrocytes which are dependent on IFNAR expression[69], however, the identity of these cues are not known. What we see in our data from the cortex is that; cells in WT mice upregulate cytokines and chemokines that seem to specifically attract CD8+ T-cells and NK-cells but not CD4+ T-cells to the brain. These cells secrete IFNγ which is known to upregulate CCR1 on microglia[70]. In accordance with this we observe CCR1 upregulation in WT but not in *Ifnar*$^{-/-}$ microglia. The CCL5 (RANTES)-CCR1 interaction is important for microglia activation[71]. Most cells of both genotypes upregulate CCL5 after infection, which may also interact with the receptor CCR5 present on microglia from both WT and *Ifnar*$^{-/-}$. However, CCL5-CCR5 does not seem to be enough to activate microglia as the majority of infected microglia in *Ifnar*$^{-/-}$ remained in a morphological non-activated state. In contrast, the uninfected WT microglia show different stages of activation, which might protect them from infection. Interestingly, the observed tropism shift and increase in Iba1-positive cellular infection by LGTV in cortex is similar to what was observed in STING-deficient mice, infected with HSV-I[72].

We observed infection in both WM and GM in *Ifnar*$^{-/-}$ brains, while infection in WT brains was mostly restricted to the latter. This restriction to GM might relate to the surprising shift in cellular tropism from neurons in WT to Iba1-positive cells in *Ifnar*$^{-/-}$ mice. Interestingly, we did not detect Iba1-positive infected cells in cerebellum nor brain stem in *Ifnar*$^{-/-}$ mice, indicating location specific differences of Iba1-positive cell populations. Interestingly, we observed an increase in the proportion of microglia/MØ in the cortex of WT mice upon

infection using snRNAseq. Sub-clustering of these cells showed a dominance of infiltrating MØ. FACS analysis confirmed the presence of infiltrating MØ (CD11b[+], CD45[high]) and resident microglia (CD11b[+], CD45[int]) in WT mice. We can only speculate why this population of microglia in WT was relatively small in the snRNAseq data set. One possibility is that activated microglia become fragile and rupture during sample processing. In *Ifnar*[-/-] mice the microglia population remained constant and very few MØ were detected. The majority of the infected Iba1-positive cells were TMEM119-positive, indicating that resident microglia are targeted by the virus. Proper activation of microglia in olfactory bulb requires IFNAR signaling in neurons and astrocytes but not in microglia[69]. In line with this, infection of cultured primary microglia from cortex showed low susceptibility to LGTV infection, independent of IFNAR expression. This discrepancy between in vitro and in vivo experimentation strongly indicates that cellular communication between neurons, astrocytes and microglia, which produce a local milieu within the brain, dictates the outcome and susceptibility of infection. Therefore, one should be cautious when extrapolating in vitro findings to the in vivo setting.

Here, we investigated the effect of the local IFN-I response on tick-borne flavivirus neurotropism. Using multimodal imaging, spanning a range from cm to nm, we were able to anatomically map viral distribution and identify target cells where we could visualize active replication and the release of virions. We found that the IFN-I response shapes viral distribution in various neuronal pathways and brain regions as well as its cellular tropism. Together, these results suggest that cell types or cell populations that belong to different brain areas may respond to viral infection in different ways, and that the response is highly region- and cell-specific. Additionally, we used snRNAseq to analyze the cellular response following infection and define the molecular mechanisms of pathogenesis. In WT mice, we found a strong IFN-I signaling and inflammatory response with IFNγ expression, infiltration of MØ, CD8+ T-cells and NK-cells, and activated microglia expressing *Ccr1*. In the absence of IFN-I signaling the cells are unable to respond to viral infection and only astrocytes showed limited IFN-independent response. The lack of IFNγ expression within the brain and absence of CCR1 on microglia may prevent their activation, thus contributing to the shift in viral tropism to resident microglia which consequently became the main target of infection. Collectively, our approach provides unprecedented insight into the outcome of viral infection, the importance of IFNs and the molecular mechanisms of viral pathogenesis. Although our study is not directly translatable to humans, it can serve as a guideline to study translational aspects of viral infection in the human brain. Future studies are required to systematically explore the temporal, spatial and cellular responses to viral infection to understand the effects of the IFN response in full, and to improve our understanding of viral neuropathogenesis.

## Methods

### Animals
C57BL/6 WT mice and *Ifnar*[-/-] mice in C57BL/6 background were kindly provided by N.O. Gekara[8]. Mice were bred as homozygotes and maintained under specific pathogen-free conditions. Animal experiments were approved by the regional Animal Research Ethics Committee of Northern Norrland and by the Swedish Board of Agriculture (ethical permits: A9-2018 and A41-2019), and all procedures were performed according to their guidelines.

### Viruses
LGTV strain TP21 was a kind gift from G. Dobler (Bundeswehr Institute of Microbiology, Munich, Germany). LGTV stock was produced in VeroB4 cells, a kind gift from G. Dobler (Bundeswehr Institute of Microbiology, Munich, Germany)[26] and harvested on day 3 post infection when cytopathic effects were apparent. Virus supernatant

was harvested and titrated on VeroB4 cells using focus-forming assay[26]. Viruses were diluted in 10-fold dilutions in DMEM supplemented with 2% FBS 100 U/mL of penicillin and 100 μg/mL streptomycin (Gibco,). Diluted viruses were then used to infect VeroB4 cells for 1 h at 37 °C and 5% $CO_2$, then inoculum was removed and replaced with avicel medium (1.2% Avicel RC-581, 2% FBS in DMEM). After 3 days of incubation avicel medium was removed and cells were fixed in 4% formaldehyde and permeabilized in PBS containing 0.5% Triton X-100 and 20 mM glycine. Viral foci were detected using primary mouse monoclonal antibodies directed against TBEV E (1786.3, 1:1000 in PBS 10% FBS[73]) and horseradish peroxidase-conjugated anti-mouse secondary antibodies (ThermoFisher 31430, 1:2000 in PBS 10% FBS), and foci were revealed by addition of TrueBlue peroxidase substrate (KPL, Gaithersburg, MD).

### Virus infection model in the mouse
Animals (7- to 13-week-old, mixed gender) were either left untreated, mock-treated (PBS) or infected with LGTV. After sedation with ketamine (100 μg/g body weight) and xylazine (5 μg/g body weight) or Isoflurane, animals were intracranially inoculated with LGTV suspended in 20 μL PBS, 1000 focus forming units for Figs. 1–3 and 10,000 for Figs. 4, 5. Infected mice were euthanized using oxygen deprivation when they developed one severe symptom, such as: >20% weight loss, bilateral eye infection, diarrhea or hind-limb paralysis; or when they developed three milder symptoms, such as: >15% weight loss, unilateral eye infection, facial edema, ruffled fur or overt bradykinesia, and/or development of stereotypies.

### Whole-mount immunohistochemistry (IHC) and OPT
Following euthanasia, cardiac perfusion was performed using 20 mL PBS followed by 20 mL 4% w/v paraformaldehyde (PFA) in PBS, after perfusion, brain was removed and further immersed in PFA for 2 h before thoroughly washed by PBS. PFA-fixed brains were fluorescently immunolabeled with anti-NS5 antibody and processed for OPT imaging[42,74]. The brain was dehydrated in a stepwise gradient of methanol (MeOH), permeabilized by repeated freeze–thawing in MeOH at −80 °C and bleached in a solution (MeOH:$H_2O_2$:DMSO at 2:3:1) to quench tissue autofluorescence. Specimens were rehydrated in TBST (50 mM Tris-HCl, pH 7.4, 150 mM NaCl, and 0.1% v/v TritonX-100), blocked with 10% v/v normal goat serum (NGS) (#CL1200-100, Nordic Biosite, Sweden), 5% v/v DMSO, and 0.01% w/v sodium azide in TBST at 37 °C for 48 h, and labeled with primary (chicken anti-NS5; Supplementary Table 1) and secondary (goat anti-chicken Alexa Fluor 680; Supplementary Table 1) antibodies diluted in blocking buffer. They were incubated at 37 °C for 4 days at each staining step. The stained tissue was mounted in 1.5% w/v low melting point SeaPlaque™ agarose (#50101, Lonza, Switzerland) and optically cleared using a BABB solution (benzyl alcohol (#1.09626.1000, Supelco, USA): benzyl benzoate (#10654752, Thermo Fisher Scientific, USA) at 1:2).

OPT imaging was performed with an in-house developed near-infrared OPT system (described in detail in Eriksson et al. 2013[74]), with a zoom factor of 1.6 or 2.0, that resulted in an isotropic voxel dimension of 16.5 μm³ and 13.2 μm³, respectively. To obtain specific fluorescent viral (NS5) signal (coupled with Alexa Fluor 680), and tissue autofluorescence signals, OPT images were acquired at the following filter settings: Ex: 665/45 nm, Em: 725/50 nm (exposure time: 7000 ms) and Ex: 425/60 nm, Em: 480 nm (exposure time: 500 ms), respectively.

To increase the signal-to-noise ratio for NS5, the pixel intensity range of all images was adjusted to display the minimum and maximum, and a contrast-limited adaptive histogram equalization (CLAHE) algorithm with a tile size of 64 × 64 was applied to projection images acquired in the NS5 channel[75]. Tomographic reconstruction was performed using NRecon software (v.1.7.0.4, Skyscan microCT, Bruker, Belgium) with additional misalignment compensation and ring artifact reduction. Image files were converted to Imaris files (.ims) using the

Imaris file converter (v9.5.1, Bitplane, UK). NS5 signal from all imaged brains was adjusted to display at min = 0, max = 200, and gamma = 1.2, and colored using red glow color scheme. The signal was super-imposed onto the corresponding tissue autofluorescence image using 3D iso-surface rendering in Imaris software (v9.5.1, Bitplane).

Of note: the variability of the NS5 signal between viral infections, together with the fact that the tissue was also used for MRI scanning and other applications after OPT acquisition, made it difficult to test the antibody penetration efficacy throughout. Still, OPT scanning displayed prominent signal in deep areas of the tissue (see Supplementary Fig. 1b), and similar staining patterns were confirmed in separate brains by confocal microscopy

## Light sheet fluorescent imaging

High-resolution images of ChP in the fourth ventricle of individual *Ifnar*$^{-/-}$ brains, previously scanned using OPT, were acquired by LSFM. To compensate for potential photobleaching effects due to homogeneous sample illumination during OPT acquisition, the sample was relabeled using both primary and secondary antibodies (see above) and cleared in BABB without agarose mounting. The brain was then scanned using an UltraMicroscope II (Miltenyi Biotec, Germany) with a 1× Olympus objective (PLAPO 2XC, Olympus, Japan), coupled to an Olympus zoom body providing 0.63–6.3× magnification with a lens-corrected dipping cap MVPLAPO 2×DC DBE objective (Olympus). For image acquisition, ChP was imaged in 3 tiles at 2.5× magnification and stitched together with 20% overlap. Left and right light sheets were merged with a 0.14 numerical aperture, which resulted in a light sheet z-thickness of 3.87 μm and 60% width, while using a 10–15 step blending dynamic focus across the field of view. Image sections were generated by Imspector Pro software (v7.0124.0, LaVision Biotec Gmbh, Germany) and stitched together using the TeraStitcher script (v9), implemented in Imspector Pro. Stitched images were then converted into Imaris files (*.ims files) using the Imaris file converter (v9.5.1, Bitplane).

## MRI acquisition

After OPT, brains were rehydrated in TBST. T1-weighted images were then acquired at 9.4 Tesla using a preclinical MR system (Bruker BioSpec 94/20, Bruker, Germany) equipped with a cryogenic RF coil (MRI CryoProbe, Bruker) running Paravision 7.0 software (Bruker). Specifically, Modified Driven Equilibrium Fourier Transform (MDEFT) sequence with 5 repetitions (TR: 3000 ms; TE: 3 ms; TI: 950 ms; voxel dimension: $40 \times 40 \times 40$ μm$^3$) was performed. Postprocessing of images involved the realignment and averaging of individual repetitions using statistical parametric mapping (SPM8) (the Wellcome Trust Centre for Neuroimaging, UCL, UK; www.fil.ion.ucl.ac.uk/spm) implemented in Matlab (R2014a, The MathWorks Inc., USA).

## Creation of OPT-MRI fusion images

OPT images with viral signal and autofluorescence signal were reconstructed in DICOM format using NRecon software (v.1.7.0.4, Bruker) followed by their conversion into NifTi using PMOD VIEW tool (v.4.2, PMOD Technologies Inc., Switzerland) or the dcm2nii tool in MRIcron software for OPT and MR images, respectively. Coregistration and normalization of OPT with the OCUM MRI template was performed using the toolbox SPMmouse in SPM8. All transformation matrices were calculated using individual tissue autofluorescence from OPT images and applied to the corresponding viral OPT images[21]. The accuracy of all transformations were assessed visually by two individual readers in the check registration tool of SPM8. Fusion images of viral OPT signal were created for each individual brain using its own MRI and with the OCUM template[21] using the PMOD VIEW tool or Amira-Avizo software (v6.3.0, Thermo Fisher Scientific) for 3D renderings. Finally, brain areas with viral signal were identified according to the OCUM atlas[21].

## Atlas-based OPT quantification

To assess the differences in viral infection between WT and *Ifnar*$^{-/-}$ mice in more detail and to provide a metric for viral infection in distinct brain regions, a Volume of Interest (VOI)-based analysis was performed using VOI analysis in the PMOD VIEW tool and the OCUM atlas, resulting in 336 automatically delineated brain VOIs. Individual VOI metrics were calculated after voxel intensities were thresholded to remove background and the highest pixel intensities were normalized to the average of the highest pixel intensities for each subject to obtain relative VOI intensities. Afterwards, all individual VOI metrics of all individual images (MOCK, WT and *Ifnar*$^{-/-}$) were normalized to the individual VOI averages of the MOCK group to obtain an arbitrary unit relative to MOCK. Thereafter, larger bilateral composite regions of interest (ROIs) were created and the arbitrary units of all individual VOIs in a composite ROI were summed to provide the total arbitrary unit in the composite ROI. Finally, unpaired t-tests were performed in GraphPad Prism (v9.3.1 GraphPad Software Inc., San Diego, CA, USA) after F-test analysis for equal variances between groups. To analyze individual VOIs within a composite ROI, multiple unpaired t-test with Holm-Sidak multiple comparison correction (alpha = 0.05) were performed.

## IHC for brain slices

Following euthanasia, cardiac perfusion with 4% PFA and PBS was performed as described above followed by immersion in 4% PFA at 4 °C overnight. PFA-fixed brain was washed in PBS, dehydrated overnight in a 30% w/v sucrose solution, snap-frozen on dry ice in Optimal Cutting Temperature medium (#361603E, VWR, USA), and stored at −80 °C until cryosectioning. The brain was sectioned along the sagittal plane at 10 μm thickness using a rotatory microtome cryostat (Microm Microtome Cryostat HM 500 M, Microm, USA). Brain sections were permeabilized and blocked in 10% v/v NGS, 0.2% v/v TritonX-100, and 1% w/v bovine serum albumin in PBS for 1 h at room temperature (RT), immunolabelled with primary and secondary antibodies (Supplementary Table 1) diluted in 2% v/v NGS and 0.5% v/v TritonX-100 in PBS for overnight at 4 °C and 1 h at RT in the dark, respectively. Confocal fluorescence microscopy was performed using a Zeiss 710 confocal microscope (Zeiss, Germany) controlled by Zeiss Zen interface (v.14.0.19.201) with Axio Observer inverted microscope equipped with Plan Apochromat 20 × 0.8, C-Apochromat 40×/1.2, and Plan Apochromat 63×/1.4 objective lens or a Leica SP8 Laser Scanning Confocal Microscope equipped with HC PL APO 20x/ 0.75 (Leica), HC PL APO 40x/1.3 or HC PL APO 63x/1.40 and Leica Application Suit X software (LAS X, v.3.5.5, Leica). For quantification of microglial infection, stitched images of cerebrum were created using a Leica SP8 Laser Scanning Confocal Microscope equipped with HC PL APO 20x/0.75 (Leica) and Leica Application Suit X software (LAS X, v.3.5.5, Leica).

## Quantification of microglial infection

Sagittal brain slices (2 *Ifnar*$^{-/-}$ and 3 WT mice, 6-8 sections per animal) were co-stained for NS5 and Iba1 and the infected microglia were quantified in Imaris (v. 9.7.2, Bitplane). Therefore, stitched confocal images of the cerebrum were converted to .ims files using the Imaris file converter (v. 9.8.0, Bitplane). After background subtraction, a colocalization channel, displaying only cells positive for both NS5 and Iba1, was built using the "coloc" tool. Then, the "cell" function was applied to the NS5 and the colocalization channel to calculate the percentage of infected microglia, using a cell background subtraction width of 3 μm. The percentage of infected microglia was calculated for all slices and expressed as mean ± standard deviation (SD). Statistical analysis was performed using an unpaired two-tailed t-test with Welch´s correction in GraphPad Prism. The level of significance was set at $p < 0.05$.

## Fixation, resin embedding, and staining of tissue for electron microscopy

ChP from the brain of LGTV-infected $Ifnar^{-/-}$ mice were prepared for electron microscopy by cardiac perfusion with 20 mL of 0.1 M phosphate buffer, followed by 20 mL of 2.5% w/v glutaraldehyde and 4% w/v PFA in 0.1 M phosphate fixative solution. Fourth ventricle ChP was dissected and further immersed in the same fixative solution for an additional 24 hours. The tissue was stained and subsequently embedded in resin using the rOTO (reduced osmium tetroxide, thiocarbohydrazide, osmium tetroxide) method[76]. The tissue samples were placed in a solution of 1.5% w/v potassium ferrocyanide and 2% w/v osmium tetroxide (OsO4), and then incubated in Pelco Biowave Pro+ (Pelco, Fresno, USA), a microwave tissue processor ("the processor"), under vacuum for 14 min. After two rinses with MilliQ water on the bench, the samples were washed twice with MilliQ water in the processor without vacuum pressurization. Then, the samples were incubated in 1% w/v thiocarbohydrazide solution for 20 min. After another MilliQ water rinse on the bench, the samples were again washed twice in the processor (no vacuum). Next, the samples were placed in 2% w/v OsO$_4$ solution and run in the processor under vacuum for 14 min, followed by an additional water and processor wash. The samples were placed in 1% w/v uranyl acetate solution and run in the processor under vacuum for 12 min, followed by another water and processor wash. The samples were then dehydrated in a stepwise ethanol gradient series:30%, 50%, 70%, 80%, 90%, 95%, and 100%, twice in the processor without vacuum. The dehydrated samples were infiltrated with an increasing concentration of Durcupan ACM resin (Sigma-Aldrich) using the following stepwise ratios of ethanol to Durcupan resin: 1:3, 1:1, and 3:1 in the processor for 3 min under vacuum. The two final infiltration steps were performed in 100% resin. Finally, the samples were transferred to tissue molds, and placed at 60 °C for 48 h for complete polymerization of the resin.

## TEM

Using a Reichert UltraCut S ultramicrotome (Leica, Germany), resin-embedded samples were trimmed, and 50 nm sections were cut using a diamond knife and placed onto copper slot grids. Resin-embedded sections were imaged using a 120 kV Talos L120C transmission electron microscope (Thermo Fischer Scientific) fitted with a LaB6 filament and Ceta 4k × 4k CMOS camera sensor. Images were acquired at 2600×, 8500×, and 36,000× magnification corresponding to a pixel size of 54.3, 17.0, and 4.1 Å/px, respectively, at the specimen level. TEM images were analyzed by ImageJ software (v. 1.8.0_172, NIH).

## Volume imaging using FIB-SEM

Resin-embedded tissue blocks were trimmed, mounted on SEM stubs, and then coated with a 5 nm platinum layer using a Q150T-ES sputter coater (Quorum Technologies, UK) before FIB-SEM volume imaging. Data was acquired using Scios Dual beam microscope (FIB-SEM) (Thermo Fischer Scientific). Electron beam imaging was acquired at 2 kV, 0.1 nA current, 1.9 × 1.9 nm pixel spacing, 7 mm working distance, 10 μs acquisition time, and 3072 × 2048 resolution using a T1 detector. SEM images were acquired every 20 nm. The working voltage of gallium ion beam was set at 30 kV, and 0.5 nA current was used for FIB milling. The specimens were imaged at 5 × 5 μm block face and 5 μm depth. FIB milling and SEM imaging were automated using the Auto Slice and View software (v4.1.1.1582, Thermo Fischer Scientific). SEM volume images were aligned and reconstructed using ImageJ (v. 1.8.0_172, NIH) with linear stack alignment, with SIFT and Multi-StackRegistration plugins[77,78]. Analysis and segmentation of SEM volume images were done using Amira-Avizo software (v2020.3.1, Thermo Fisher Scientific).

## Single nuclei isolation and sequencing

After euthanasia (day 5 for WT and day 3 for $Ifnar^{-/-}$), brains were collected in cold Hibernate A GlutaMAX medium (Thermo Fisher) without cardiac perfusion. For each condition one male and one female mouse were pooled and single nuclei were isolated from the cerebral cortex using the protocol modified from Gaublomme et al. 2019[79]. Cerebral cortexes from the hemisphere that did not receive needle injection from one male and one female mouse were manually dissected and homogenized in lysis buffer (10 mM Tris pH 7.4, 146 mM NaCl, 1 mM CaCl2, 21 mM MaCl2, 0.1% NP40 and 40 U/ml RNase inhibitor) using Dounce homogenizer for 10-20 times. Brain homogenates were washed with wash buffer (10 mM Tris pH 7.4, 146 mM NaCl, 1 mM CaCl2, 21 mM MaCl2, 0.01% BSA and 40 U/ml RNase inhibitor) and filtered through 20 μm pre-separation filters (Miltenyi Biotec, 130-041-407) and were centrifuged with a swing bucket at 500xg, 4 °C for 5 min. The supernatant was removed and resuspended in the stain buffer (10 mM Tris pH 7.4, 146 mM NaCl, 1 mM CaCl$_2$, 21 mM MaCl$_2$, 2% BSA and 40 U/ml RNase inhibitor). Nuclei were stained with 10 μg/ml DAPI and subjected to sorting from cell debris using FACS (BD FACSAria III). The gate was set to sort for DAPI positive population (FSC 302, SSC 313 and BV421 313(log)). The flow rate was adjusted to maintain about 10 nl/nuclei and with 80% efficiency. Sorted nuclei were transferred into low binding tubes and centrifuged using a swing bucket at 500xg, 4 °C for 5 min. The overall quality of sorted nuclei was assessed under a microscope. Overall, 80% viable nuclei were then diluted for snRNAseq processing, using 10x Genomics 3' v2 kit according to manufacturer's instructions. The target recovery cells for WT and $Ifnar^{-/-}$ conditions were 8800 cells. Reverse transcription, cDNA amplification and library construction were performed, and libraries were sequenced on an Illumina 6000 platform according to the manufacturer's instruction.

## Data processing and quality control snRNAseq

A custom reference genome was made by combining cellranger-arc-GRCh38-2020-A-2.0.0, NC_003690 (LGTV), and the neomycin gene sequence from pMC1neo (Addgene https://www.addgene.org/vector-database/3549/), from XhoI to BamHI, based on design in Müller et al.[8] Demultiplexed reads were processed using CellRanger v6.1.2 with options expect-cells = 8000. All data sets were filtered prior to further processing. First, genes detected in fewer than 3 cells or nuclei with <300 features were removed. A threshold of <10% mitochondrial reads were applied and reads for the lncRNA Malat1 were removed. Potential doublets were addressed by (i) removing cells with high number (>5000) of detected features, (ii) removing nuclei with co-expression of female and male genes (Xist and Eif2s3y) and (iii) following cell annotation and subclustering of large cell types, removing distinct clusters showing co-expression of multiple cell type specific markers (e.g. neuronal and glial markers). The relative expression levels of the Neo gene present in $Ifnar^{-/-}$[8] and LGTV, were used to quantitatively verify the sample identities.

## Cell annotation

Data analysis was performed in R (v.4.2.1, The R Foundation) through RStudio Desktop (v.2022.07.1, RStudio, PBC), using the Seurat package (v 4.1.0)[80]. Data were normalized (NormalizeData), top 2000 variable features identified for each data set (FindVariableFeatures) and data integrated using FindIntegrationAnchors and IntegrateData in Seurat. The integrated expression matrix was scaled (ScaleData), subjected to principal component analysis (RunPCR) and the first 30 dimensions used as input for RunUMAP and to construct a shared nearest neighbor graph (SNN, FindNeighbors). Clustering was performed using FindClusters at resolution 1. Following data integration and dimensional reduction/clustering, gene expression data (assay "RNA") was scaled and centered (ScaleData). The positive differential expression of each other cluster compared to all others (FindAllMarkers, test

"wilcox") in combination with expression of established marker genes[81,82], Allen Brain Atlas, https://portal.brain-map.org/) was used to assign preliminary identities (excitatory neurons; neuron.ex, inhibitory neurons; neuron.in, microglia/macrophages; micro/MØ, astrocytes; astro, oligodendrocytes; oligo, oligodendrocyte progenitor cells; OPC, vascular leptomeningeal cells; VLMCs, pericytes; peri, endothelial cells; endo, CD8+ T-cells and NK-cells; CD8+ NK, choroid plexus epithelial cells; ChP). Low-quality cells were identified as clusters with a combination of (i) few significant differentially expressed genes and (ii) high fraction of transcripts from mitochondrial and ribosomal genes and removed manually. To distinguish between microglia and other infiltrating immune cell population, this population was subsetted and the steps for dimension reduction, clustering and differential gene expression repeated.

### DEG analysis, GSEA and CellChat
Differentially expressed genes were identified using FindMarkers in Seurat, test MAST, applying a threshold of $\log_2$ fold change ($\log_2$FC) >1 and adjusted $p$ values ($p_{adj}$) <0.05. GSEA was performed using the fgsea package version 1.22.0 in R[83]. Genes were ordered by average $\log_2$FC on the MSigDB Reactome pathways obtained using the misgdbr package version 7.5.1. Pathways with a Benjamini-Hochberg corrected $p$-value <0.05 were considered significant. A comparison of mock (PBS) treated samples with untreated samples showed little transcriptional response to the injection procedure as such. Therefore, all further DEG analysis and GSEA was made comparing infected animals with uninfected (mock and untreated). Cell-cell communication within infected data sets was analyzed by CellChat version 1.6.1[41]. A cutoff of 25% gene expression within the group was applied and only cell types with more than 10 cells were included. To specifically characterize communications involving microglia, cells annotated as MØ were separated from the micro/MØ cluster prior to analysis.

### Isolation and infection of primary cells
Mixed glial cells were isolated from mice between postnatal day 1 and 4. Cerebral cortices were isolated and titrutated in HBSS containing 0.4% glucose 100 U/mL of penicillin and 100 µg/mL streptomycin. Then cells were seeded in poly-D-lysine (Sigma, P0899) coated T75 tissue culture flasks. Cells were cultured in DMEM supplemented with 10% heat inactivated FBS and 100 U/mL of penicillin and 100 µg/mL streptomycin. Upon reaching confluency microglial cells were isolated from the mixed glial cell culture by gentle trypsinization (trypsinization in 0.05% trypsin, 200 µM EDTA, 20% HBSS and 80% DMEM) for 25 minutes[84]. The isolated, detached microglia were then grown in DMEM, supplemented with 10% FBS, 100 U/mL of penicillin and 100 µg/mL streptomycin (Gibco), and mixed glial-conditioned medium (1:1). Monolayers of microglia were infected with LGTV (MOI 1) for 1 h at 37 °C and 5% $CO_2$ before the inoculum was removed and replaced with fresh medium. Cell supernatant was harvested at 72 h post infection and viral titers determined by focus forming assay[26], and normalized to input control 3 hours post infection.

### RNA extraction and RT-qPCR
Total RNA was extracted from cerebral cortexes from one hemisphere of infected mice and viral replication quantified by qPCR[9]. In short, the cerebral cortex was manually dissected and homogenized in QIAzol Lysis Reagent (Quiagen) using 1.3 mm Chrome-Steel Beads (BioSpec) and the tissue homogenizer FastPrep-24 (MP). The RNA was extracted using the Nucleo-Spin RNA II kit (Macherey-Nagel). 1000 ng of total RNA was used as input for cDNA synthesis using High-capacity cDNA Reverse Transcription kit (Thermo Fisher). LGTV RNA was quantified using qPCRBIO probe mix Hi-ROX (PCR Biosystems) and primers recognizing NS3, forward primer 5′-AACGGAGCCATAGCCAGTGA-3′, reverse primer 5′-AACCCGTCCCGCCAC TC-3′ and probe FAM-AGAGACAGATCCCTGATGG-MGB, with a sensitivity of 10 copies[15]. GAPDH was used as housekeeping gene, detected by QuantiTect primer assay (QT01658692, Qiagen) and the qPCRBIO SyGreen mix Hi-ROX (PCR Biosystems). All experiments were run on a StepOnePlus real-time PCR system (Applied Biosystems).

### Flow cytometry
After euthanasia (day 5 for WT and day 3 for $Ifnar^{-/-}$) mice were perfused with 15 mL cold PBS and the cerebral cortex collected in cold Hank's Balanced Salt Solution (HBSS). The tissue was homogenized through a 70 µm cell strainer, digested with a collagenase solution (PBS, 500 µg/mL collagenase D, 10 µg/mL trypsin inhibitor TLCK, 10 µg/mL DNase I, 10 mM HEPES) for 20 min at 37 °C. Cells were washed in FACS buffer (PBS, 2% FBS, 2 mM EDTA), centrifuged at 1200 rpm, 5 min at 4 °C and purified on a discontinuous 70 to 30% Percoll gradient by centrifugation at 800 xg for 20 min at room temperature. Cells were stained with directly conjugated antibodies (Supplementary Table 1) diluted in FACS buffer for 30 min on ice, washed twice in FACS buffer and analyzed on a ZE5 Cell Analyzer (BioRad) with Everest software (v.3.1.18.0). Analysis was performed using FlowJo software (v10.0.7r2, BD).

### Reporting summary
Further information on research design is available in the Nature Portfolio Reporting Summary linked to this article.

## Data availability
All data generated or analyzed during this study are provided in the Supplementary Information/Source Data file. Raw single nuclei RNA-seq data generated in this study have been deposited in the ArrayExpress database under accession code E-MTAB-12131. LGTV strain TP21 viral genome sequence is publicly accessible in GenBank (NC_003690). Raw image data can be requested from the corresponding authors with reasonable means to transfer large data files. Source data are provided with this paper as a Source Data file. Source data are provided with this paper.

## Code availability
The scripts are available at https://github.com/ERosendal/LGTV_WT_Ifnar_10x.

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

## Acknowledgements

We thank G. Dobler (Bundeswehr Institute of Microbiology, Munich, Germany) for providing a stock of LGTV strain TP21 and VeroB4 cells; and N.O. Gekara (Department of Molecular Biosciences, Stockholm University, Stockholm, Sweden) for providing the *Ifnar*$^{-/-}$ mouse colony. We thank M. Eriksson and C. Nord (UCMM, Umeå University, Umeå, Sweden) for OPT training, and J. Gilthorpe (Department of Integrative Medical Biology, Umeå University, Umeå, Sweden) for invaluable discussions. Grants: MIMS Excellence by Choice Postdoctoral Program under the patronage of Emmanuelle Charpentier from the Kempe Foundation SMK-1532, ES; and Knut and Alice Wallenberg Foundation KAW2015.0284, NC. Wallenberg Centre for Molecular Medicine Umeå, LAC; Umeå University Industrial Doctoral School, ISM; Umeå University Medical Faculty, UA, DM; The Kempe foundation, UA; The Laboratory for Molecular Infection Medicine Sweden (MIMS), AKÖ; The Swedish Research Council grants 2018-05851, LAC; 2017-01307, JH; 2017-01307, UA; 2020-06224, AKÖ; and 2018-05851, AKÖ.

We also acknowledge Umeå Center for Microbial Research (UCMR); the Biochemical Imaging Center at Umeå University (BICU), and the National Microscopy Infrastructure for microscopy support (NMI; VR-RFI 2019-00217); the Umeå Centre for Electron Microscopy (UCEM), a Sci-LifeLab National Cryo-EM facility for EM support (VR-RFI 2016-00968); the Small Animal Research and Imaging Facility (SARIF), and the Flow@CliMi platform at Umeå University for their flow cytometry support; and the Single Cell Core Facility @ Flemingsberg campus (SICOF), Karolinska Institute for their technical advice. The computations were performed using resources provided by the Swedish National Infrastructure for Computing (SNIC) through the Uppsala Multidisciplinary

Center for Advanced Computational Science (UPPMAX) under project SNIC 2022/5-18. Illustration (Figs. 1a, 2b and 5a) were made with BioRender.com.

## Author contributions

N.C., E.R., S.M.A.W., E.S., J.Z., L.A.C., U.A., D.M., A.K.Ö. designed the experiment, N.C., S.M.A.W., E.R. and E.N. performed animal experiments, N.C., S.M.A.W., M.H. and E.N. acquired images and analyzed data from OPT, LSFM and confocal microscopy, S.M.A.W., F.M. and D.M. acquired images and analyzed data from MRI and E.S., J.Z. and L.A.C. acquired images and analyzed data from E.M. RL isolation, infection and analysis of microglia in vitro. N.C., J.H., I.M.S. processed samples for snRNAseq, E.R., J.H. and G.E. analyzed snRNAseq data. E.R. processed samples, ran the flow cytometry and analyzed the data. N.C. and A.K.Ö. generated final figures. L.A.C., U.A., D.M., J.H. and A.K.Ö. supervised the experiments. N.C., E.R., S.M.A.W., E.S., L.A.C., U.A., D.M., A.K.Ö., wrote and edited the manuscript. All authors revised the manuscript.

## Funding

## Competing interests

The authors declare no competing interests.

## Additional information

**Peer review information** : *Nature Communications* thanks Elizabeth MC Hillman and the other, anonymous, reviewer(s) for their contribution to the peer review of this work. Peer reviewer reports are available.

