## [Peer Review File · Nature Communications]

Type I interferon shapes brain distribution and tropism of tick-borne flavivirusReviewers' Comments:

Reviewer #1:

Remarks to the Author:

Chotiwan and colleagues addressed the question what impact type I interferon receptor (IFNAR) signaling has on the dissemination and disease development of intracranially tick-borne encephalitis virus (TBEV) injected mice. By application of multimodal imaging and single nuclei RNA sequencing the authors obtained data that supported the conclusion that in wild type mice the infection was limited to gray matter of the olfactory, visual and somatosensory systems, whereas in IFNAR deficient mice the virus was also detected in the white matter, meninges and choroid plexus. Interestingly, the cellular tropism of the virus shifted from neurons in wild type mice to myeloid cells in IFNAR deficient mice.

In order to localize virus infection within the brain, wild type and IFNAR deficient mice were intracranially infected with TBEV and 7 and 4-5 days later, respectively, the brains were analyzed by optical projection tomography (OPT) and ex vivo magnetic resonance imaging (MRI). This approach allowed specific identification of differences in the virus localization in wild type and IFNAR deficient mice with high spatial resolution. However, the shortcoming of this approach is that different time points were analyzed. Instead, analysis of a series of time points, some of which should be identical for wild type and IFNAR deficient mice would have facilitated drawing conclusions.

The authors highlighted that in WT brains weak viral signal in the meninges and the interior wall of third ventricle were detected by OPT, whereas confocal analysis did not prove the presence of viral antigen in these regions. In contrast, IFNAR deficient brains showed intense viral OPT signal within meninges that was confirmed by confocal microscopy. The authors concluded that in wild type brains some non-specific antibody trapping took place in the meninges and the interior wall of the third ventricle. The information that is relevant to draw these conclusions currently is spread in several figures and supplementary figures, which makes it rather difficult to follow the argument. Therefore, it is recommended to show the relevant data in the main figures and not to hide them somewhere. The authors were able to prove strengths and weaknesses of the combined OPT MRI visualization approach and they should provide the corresponding information in an easily accessible manner.

In a next step, the authors used transmission electron microscopy (TEM) to study choroid plexus epithelial cells on further enhanced resolution. These experiments revealed massive distortion of endoplasmic reticulum membranes in infected cells. In higher resolution images even formation of viral replication complexes could be demonstrated. Thus, the current study dissects conditions of TBEV infection of the brain down to subcellular conditions.

The following chapter provides single nucleus RNA sequencing data. Conceptually, this part does not add significant new information to the study. Although the authors undertook some effort to verify snRNA-seq data on the protein level, e.g. by confocal imaging in Fig. 5f, additional experiments such as FACS analysis of infiltrating cells is needed. In the end, Figures 4 and 5 do not contribute significant new information, as also highlighted by the more technically motivated titles of the corresponding Figure legends. This raises the question of whether this part of the study is needed to support the major conclusions of the study. In my opinion the study would profit from focusing on the anatomical studies and omitting the snRNA-seq data.

Reviewer #2:

Remarks to the Author:

The manuscript by Chotiwan et al utilizes a novel approach of Optical Projection Tomography (OPT) combined with MRI imaging to examine the location of Langkat Virus (LGTV) in wildtype (WT) and Interferon receptor 1 deficient (KO) mice. The authors show more widespread virus in the KO mice

than the WT mice, which is generally expected with KO mice. They further analyze tropism and show that KO mice have additional infection of microglia cells. Single cell analysis shows there is a heightened type I IFN response in the KO mice, which the authors suggest may be due to the presence of IFN gamma producing CD8+ NK cells, which are only observed in the KO mice. Although most of the results are not unexpected in this study, the combined technology utilized is novel and highly useful. However, there are some major concerns that should be addressed:

1) Lines 97-100 indicate that there were some discrepancies between the OPT signal and staining by confocal microscopy in that some positive areas by OPT were not positive by confocal. This is not unexpected and is useful information. Does the reverse also occur, where there areas of staining by confocal that was not positive by OPT? A direct comparison of specific regions of brain (not sectioning through the entire brain, but of areas known for LGTV infection) would help really compare these two measurements. Is OPT more sensitive in some areas and less in others (ie – is staining in the ventricle more sensitive, but maybe less so in the hippocampus or other areas)? The intensity staining difference between WT and KO mice (Sup Fig 1), despite similar levels of virus in the CNS (Supp fig 4b) suggest that there may be differences in sensitivity in different areas of tissue.

2) The authors repeatedly say there is a shift in LGTV infection in IFNAR KO mice from neurons to microglia. But is this a shift in infection or an additional set of infected cells? It would be very useful to know if the relative number of infected neurons changed between WT and KO mice – ie was this a shift in what cells are infected or is it an additional population of cells infected.

3) The switch in microglia population and macrophage population in Fig. 5d is quite shocking. What are the parameters of the mapping (this is not clearly described anywhere in the paper) and is the presence of the macrophage population indicative of infiltration or a shift in gene expression of the key genes used to cluster the cell populations? If it is infiltrating macrophages, did the microglia population die off or were the cells too fragile for sorting? This could easily be examined by completing a basic flow analysis on infected WT versus KO mice for CD45, CD11b, Ly6C, and a few other markers. This would also allow the authors to more closely look at the potential of the CD8 NK cell population rather than relying only on single cell data.

Reviewer #3:

Remarks to the Author:

This manuscript presents a multimodal imaging-transcriptomics approach to investigate viral neuropathogenesis. Using OPT, they perform tomographic reconstruction to enable whole brain visualization of viral distribution. After OPT acquisition, the authors acquire ex vivo MR images to co-register with viral OPT signal for extraction of anatomical information. This is then followed by light sheet and electron microscopy to achieve cellular and subcellular viral localization respectively. Finally, the authors utilize single cell RNA sequencing to assess the cellular response induced by viral brain infection. This comprehensive pipeline allows the authors to characterize the anatomical distribution and cellular response of LGTV infection in the presence and absence of IFNAR signaling.

The work presented is comprehensive and interesting. In the opinion of this reviewer, the main strength of the manuscript lies in the imaging-transcriptomics pipeline that the authors have developed to investigate viral brain infection, and this pipeline should perhaps be considered the main focus of the manuscript. In contrast, the biological findings presented are primarily correlative and appear to have several major limitations (discussed below). As such, the manuscript might benefit from reframing to better emphasize the technical aspects of the work – elaborating on how the novel approaches used offer advantages over historical investigations. In turn, the shortcomings of biological conclusions and necessary follow-up experiments (to prove causation) should be better discussed.

Below is a detailed list of major and minor concerns:

Major:

1. Although the authors state that they perform intracranial inoculation of the virus to bypass the BBB, this is very concerning for a paper investigating the tropism of a flavivirus. The route of inoculation likely impacts viral distribution and cellular response. Increasing evidence suggests the role of brain endothelial cells as active players in the CNS immune response. Moreover, this approach means that these mice have not been allowed to mount a preceding peripheral immune response, and therefore the local milieu is highly artificial. This limitation of the study needs more attention, otherwise, it is impossible to understand the relevance of the authors' findings. Can the authors elaborate on why this inoculation route was chosen?

2. There are anatomical differences between rodents and humans that should also be addressed. The authors emphasize that LGTV infection in IFNAR^{-/-} brains is highly localized to the rostral migratory stream. However, the presence of the RMS in the adult human brain is controversial. This brings into question the translational relevance of the authors' findings.

3. MRI for atlasing OPT data: The authors take great care justify each step of their experiment, however it does not seem that they leverage the full potential of the technologies used. For example, atlas registration would allow analysis of OPT data in a much more quantitative way, for example by giving a metric of infection for every atlas-defined region.

4. Whole-mount IHC staining with 4 days of incubation time sounds short. Complete, homogenous penetration of labels might not have been achieved. Some of the imaging data indicates this, with stronger signals towards the surface. What steps were taken to ensure uniform penetration of labels? Is it possible that the OPT tomographic images show heterogenous / superficial labeling? Could OPT images also be shown in cross section?

5. Controls / mock group brain infection OPT data was not noticeably different from infected WT brains as shown in Suppl Fig 1. Moreover, the infected WT data presented in Fig 1 of the main text does not look representative of its cohort. Can the authors explain this?

6. The main text does not clearly state how many specimens were used for either experiment. Some information can be found in the captions, but for most experiments the N remains unclear. In some, the N also seems worryingly small and needs discussion. Please clarify.

7. The identification of CD8⁺ NK cell infiltration and upregulated IFN γ expression in IFNAR^{-/-} but not WT brains is interesting. However, without adoptive transfer or ablation studies the authors cannot conclude whether, in the absence of IFNAR signaling, this response is maladaptive or protective. The molecular mechanisms are weak.

Minor:

1. Line 92: The authors cite work that is currently 'in preparation'. This makes it challenging to assess the validity and rigor of the OCUM brain template.

2. Claim in line 96: "IFN-I response protects meninges and ChP from viral infection". This conclusion cannot be derived directly from the experiments / data shown. There is an indication that this is true, but causality has not been proven.

3. Line 487: Microglia quantification. Analyzing only 6 slices provides hardly sufficient statistical power to formulate a valid statement. Also, how were the slices chosen to be anatomically equivalent such that they can be compared directly?

4. Co-registration of OPT with MRI: The authors claim good alignment - how was this measured?

5. Line 424: "To increase the signal-to-noise ratio for NS5, the pixel intensity range of all images was adjusted to display the minimum and maximum, and a contrast-limited adaptive histogram equalization (CLAHE) algorithm with a tile size of 64 x 64 was applied to projection images acquired in the NS5 channel."

Contrast enhancement does not increase SNR, so this statement should be corrected. Also, CLAHE boosts contrast locally, but risks enhancing background signal in areas absent of signal. The authors should better explain their reasoning for choosing these post-processing steps and ensure that they are not impacting / altering their conclusions.

6. Line 436: What are the "phototoxic effects"?

7. Line 542: Inconsistency → euthanasia for Ifnar is stated to have occurred after 3 days, instead of 4-5 days as stated in main text and WT 5 instead of 6-7

REVIEWER COMMENTS

We are very grateful for the reviewers' comments to the manuscript and the additional experiments suggested. As indicated by reviewers #1 and #2, we performed FACS analysis of infiltrating cells, and we found important discrepancies between the snRNAseq data and the newly acquired FACS data. To clarify these discrepancies, we performed several new control experiments; qPCR of Cd8 and Ifng, and confocal microscopy to detect infiltrating CD45 cells. These experiments indicated that the WT and Ifnar^{-/-} snRNAseq samples may have been swapped during our original analysis.

To reach a definitive answer we checked the identity of the snRNAseq samples by confirmation of the neomycin marker gene for the Ifnar^{-/-} as well as viral read (LGTV genome) for infection. This way, we could confirm without any doubt that the snRNAseq samples had been mislabeled during the original analysis, WT mice should have been Ifnar^{-/-} and vice versa. We have now reanalyzed the complete snRNA seq data and produced new figures for the manuscript.

The message of the manuscript has changed substantially, in a good way. The snRNA-seq results are now more consistent with the imaging. In our revised manuscript we have extensively evaluated the putative mechanism for the tropism shift towards microglia in Ifnar^{-/-} mice and have further developed atlas-based quantification of viral fluorescence signal as requested by reviewer #3.

Reviewer #1 (Remarks to the Author):

Chotiwan and colleagues addressed the question what impact type I interferon receptor (IFNAR) signaling has on the dissemination and disease development of intracranially tick-borne encephalitis virus (TBEV) injected mice. By application of multimodal imaging and single nuclei RNA sequencing the authors obtained data that supported the conclusion that in wild type mice the infection was limited to gray matter of the olfactory, visual and somatosensory systems, whereas in IFNAR deficient mice the virus was also detected in the white matter, meninges and choroid plexus. Interestingly, the cellular tropism of the virus shifted from neurons in wild type mice to myeloid cells in IFNAR deficient mice.

In order to localize virus infection within the brain, wild type and IFNAR deficient mice were intracranially infected with TBEV and 7 and 4-5 days later, respectively, the brains were analyzed by optical projection tomography (OPT) and ex vivo magnetic resonance imaging (MRI). This approach allowed specific identification of differences in the virus localization in wild type and IFNAR deficient mice with high spatial resolution. However, the shortcoming of this approach is that different time points were analyzed. Instead, analysis of a series of time points, some of which should be identical for wild type and IFNAR deficient mice would have facilitated drawing conclusions.

We thank the reviewer for this comment; however, the scope of this manuscript was not to compare the disease course between the genotypes but to look at the viral distribution and tropism at clinical endpoint of infection. We also analyzed the viral load in the cortex at endpoint (supplementary fig 4c) and the viral load was similar between the different genotypes.

The authors highlighted that in WT brains weak viral signal in the meninges and the interior wall of third ventricle were detected by OPT, whereas confocal analysis did not prove the presence of viral antigen in these regions. In contrast, IFNAR deficient brains showed intense viral OPT signal within meninges that was confirmed by confocal microscopy. The authors concluded that in wild type brains some non-specific antibody trapping took place in the meninges and the interior wall of the third

ventricle. The information that is relevant to draw these conclusions currently is spread in several figures and supplementary figures, which makes it rather difficult to follow the argument. Therefore, it is recommended to show the relevant data in the main figures and not to hide them somewhere. The authors were able to prove strengths and weaknesses of the combined OPT MRI visualization approach and they should provide the corresponding information in an easily accessible manner.

Thank you for pointing this out, we have now moved supplementary fig 2 into main fig 2b.

In a next step, the authors used transmission electron microscopy (TEM) to study choroid plexus epithelial cells on further enhanced resolution. These experiments revealed massive distortion of endoplasmic reticulum membranes in infected cells. In higher resolution images even formation of viral replication complexes could be demonstrated. Thus, the current study dissects conditions of TBEV infection of the brain down to subcellular conditions.

The following chapter provides single nucleus RNA sequencing data. Conceptually, this part does not add significant new information to the study. Although the authors undertook some effort to verify snRNA-seq data on the protein level, e.g. by confocal imaging in Fig. 5f, additional experiments such as FACS analysis of infiltrating cells is needed. In the end, Figures 4 and 5 do not contribute significant new information, as also highlighted by the more technically motivated titles of the corresponding Figure legends. This raises the question of whether this part of the study is needed to support the major conclusions of the study. In my opinion the study would profit from focusing on the anatomical studies and omitting the snRNA-seq data.

*We thank the reviewer for the suggestion, and we performed additional FACS analysis to confirm the snRNAseq data. After the FACS analysis, we realized that the results were not in line with the snRNAseq analysis, and after several control experiments we found that the datasets from snRNAseq were switched so that WT was actually *Ifnar*^{-/-} and vice versa. We have now reanalyzed the complete snRNAseq data and show that the inflammatory milieu in WT, with infiltration of CD8 T cells and IFN γ expression, results in *Ccr1* upregulation on microglia and activation of the WT microglia. In *Ifnar*^{-/-} mice on the other hand, we lack CD8 T cell infiltration, *Ifng* expression and activation of the microglia. We propose this as one of the reasons for the cellular tropism shift observed in *Ifnar*^{-/-} mice. We are grateful to the reviewer for this suggestion, which highlighted a swap in the analysis and think the snRNAseq analysis well complement the anatomical studies.*

We have changed figure title 5 to "Single nuclei RNA sequencing (snRNAseq) analysis of LGTV infection show strong IFN response in the cortex of WT mice." and figure title 6 is changed into "IFN signaling is important for infiltration of immune cells and activation of microglia after LGTV infection."

Reviewer #2 (Remarks to the Author):

The manuscript by Chotiwan et al utilizes a novel approach of Optical Projection Tomography (OPT) combined with MRI imaging to examine the location of Langkat Virus (LGTV) in wildtype (WT) and Interferon receptor 1 deficient (KO) mice. The authors show more widespread virus in the KO mice than the WT mice, which is generally expected with KO mice. They further analyze tropism and show that KO mice have additional infection of microglia cells. Single cell analysis shows there is a heightened type I IFN response in the KO mice, which the authors suggest may be due to the

presence of IFN gamma producing CD8+ NK cells, which are only observed in the KO mice. Although most of the results are not unexpected in this study, the combined technology utilized is novel and highly useful. However, there are some major concerns that should be addressed:

1) Lines 97-100 indicate that there were some discrepancies between the OPT signal and staining by confocal microscopy in that some positive areas by OPT were not positive by confocal. This is not unexpected and is useful information. Does the reverse also occur, where there are areas of staining by confocal that was not positive by OPT? A direct comparison of specific regions of brain (not sectioning through the entire brain, but of areas known for LGTV infection) would help really compare these two measurements. Is OPT more sensitive in some areas and less in others (ie – is staining in the ventricle more sensitive, but maybe less so in the hippocampus or other areas)? The intensity staining difference between WT and KO mice (Sup Fig 1), despite similar levels of virus in the CNS (Supp fig 4b) suggest that there may be differences in sensitivity in different areas of tissue.

We thank the reviewer for the comment. There are limitations with the OPT protocol, although it provides high resolution, single infected cells scattered might not be detected. The method works best for detecting clusters of infected cells. Penetration of the antibodies into the brain may also affect the signal. To address this question, we have now added a supplementary fig 1b which shows the penetration of our antibody in OPT cross sections.

Although, it has been reported that hippocampus is infected by LGTV in IRF7^{-/-} mice we were not able to detect viral signal in composite hippocampus of WT or Ifnar^{-/-} mice. With quantification however, one single region of hippocampus showed infection in WT (shown in supplementary fig 2c-d).

2) The authors repeatedly say there is a shift in LGTV infection in IFNAR KO mice from neurons to microglia. But is this a shift in infection or an additional set of infected cells? It would be very useful to know if the relative number of infected neurons changed between WT and KO mice – ie was this a shift in what cells are infected or is it an additional population of cells infected.

This is a good suggestion, and to address this we looked at the absolute number of infected cells in the slides and did not see a statistical difference between WT and Ifnar^{-/-} mice. We have also added an alignment to the LGTV genome in our snRNAseq data. In fig 5e, we now show that the cells with the highest number of LGTV reads in WT are neurons and pericytes while in Ifnar^{-/-} it is microglia and ChP cells. These data suggest that there is a shift in infected cells and not just an additional set of cells.

3) The switch in microglia population and macrophage population in Fig. 5d is quite shocking. What are the parameters of the mapping (this is not clearly described anywhere in the paper) and is the presence of the macrophage population indicative of infiltration or a shift in gene expression of the key genes used to cluster the cell populations? If it is infiltrating macrophages, did the microglia population die off or were the cells too fragile for sorting? This could easily be examined by completing a basic flow analysis on infected WT versus KO mice for CD45, CD11b, Ly6C, and a few other markers. This would also allow the authors to more closely look at the potential of the CD8 NK cell population rather than relying only on single cell data.

We thank the reviewer for these suggestions and during the revision and FACS analysis we realized that the snRNA datasets were swapped, WT was Ifnar^{-/-} and vice versa. We have now performed extensive reanalysis of all the data and added several controls, supplementary fig 4a, d and fig. 6e .g.

Details regarding the parameters for annotation of macrophages versus microglia have been added to the manuscript and supplementary fig 5b. With FACS, we now confirm infiltration of macrophages specifically in infected WT mice, added as fig 6e and we also show infiltrating CD45+ cells after infection of WT mice with confocal microscopy, fig 6g. We see the microglial population with FACS analysis in WT mice and the most likely explanation why this population is reduced in the snRNAseq is that the activation makes them fragile and they are lost during nuclei isolation. We also looked at the CD8/NK population with FACS and detected infiltration of distinct populations of CD335+ and CD8+ cells in infected WT mice, supplementary fig 5a.

Reviewer #3 (Remarks to the Author):

This manuscript presents a multimodal imaging-transcriptomics approach to investigate viral neuropathogenesis. Using OPT, they perform tomographic reconstruction to enable whole brain visualization of viral distribution. After OPT acquisition, the authors acquire ex vivo MR images to co-register with viral OPT signal for extraction of anatomical information. This is then followed by light sheet and electron microscopy to achieve cellular and subcellular viral localization respectively. Finally, the authors utilize single cell RNA sequencing to assess the cellular response induced by viral brain infection. This comprehensive pipeline allows the authors to characterize the anatomical distribution and cellular response of LGTV infection in the presence and absence of IFNAR signaling.

The work presented is comprehensive and interesting. In the opinion of this reviewer, the main strength of the manuscript lies in the imaging-transcriptomics pipeline that the authors have developed to investigate viral brain infection, and this pipeline should perhaps be considered the main focus of the manuscript. In contrast, the biological findings presented are primarily correlative and appear to have several major limitations (discussed below). As such, the manuscript might benefit from reframing to better emphasize the technical aspects of the work – elaborating on how the novel approaches used offer advantages over historical investigations. In turn, the shortcomings of biological conclusions and necessary follow-up experiments (to prove causation) should be better discussed.

We thank the reviewer to see the potential of our work. After several considerations and careful revision of our data according to suggestions by the reviewer 1 and 2, we found that the transcriptomics results support the observations by imaging techniques and sheds further light into the molecular mechanism of viral pathogenesis; thus bringing significant interest to the field of virology as well. Moreover, we have another manuscript explaining the methodology of MRI template creation and OPT registration in the pipeline (Willekens et al. bioRxiv doi: <https://doi.org/10.1101/2022.11.14.516420>), where we included a discussion on technical aspects of our method.

Below is a detailed list of major and minor concerns:

Major:

1. Although the authors state that they perform intracranial inoculation of the virus to bypass the BBB, this is very concerning for a paper investigating the tropism of a flavivirus. The route of inoculation likely impacts viral distribution and cellular response. Increasing evidence suggests the role of brain endothelial cells as active players in the CNS immune response. Moreover, this approach means that these mice have not been allowed to mount a preceding peripheral immune response,

and therefore the local milieu is highly artificial. This limitation of the study needs more attention, otherwise, it is impossible to understand the relevance of the authors' findings. Can the authors elaborate on why this inoculation route was chosen?

*We understand the reviewers concern about bypassing the BBB, however, Langat virus (a BSL2 pathogen) which is a model for tick-borne encephalitis virus (a BSL4 pathogen in USA and BSL3 pathogen in Europe), is not pathogenic enough to use peripheral infection routes in WT mice because the viral load in the brain is too low. The infection in *Ifnar^{-/-}* mice on the other hand, is too fast and the mice die 72 h after peripheral infection with uncontrolled viremia in all peripheral organs and very low viral levels in the brain. Our objective is to understand tropism in the brain within a controlled setting, with equal dose of the virus reaching the brain and through that investigate the importance of IFNAR specifically in the brain. In addition, we also developed a new technological platform to study viral infection and tropism in whole brain. We agree that the route of inoculation is very important but that question needs to be addressed with tick-borne encephalitis virus and is outside the scope of this manuscript.*

2. There are anatomical differences between rodents and humans that should also be addressed. The authors emphasize that LGTV infection in IFNAR^{-/-} brains is highly localized to the rostral migratory stream. However, the presence of the RMS in the adult human brain is controversial. This brings into question the translational relevance of the authors' findings.

It is completely true that the existence of the rostral migratory stream in humans is controversial, since the anatomy is quite different in the rodent brain compared to humans (Curtis, M. A. et al Science 2007). Researchers have been able to detect immature neurons in the infant human SVZ and RMS which were nearly extinct by adulthood (Senai N. et al. Nature 2011 Sep 28;478(7369):382-69). However, Wang et al. (Cell Res. 2011 Nov;21(11):1534-50) have provided evidence that neuroblasts continue to exist in the RMS of the adult human brain. Even though the existence of the RMS in the adult human brain is debatable, the olfactory bulbs in humans, albeit very small compared to mice, contain similar cell layers which project to similar cortical olfactory areas. Many of these olfactory areas were found infected in mice with LGTV, however, we have not claimed that our results are translational. We think however, that our results could be used to design and study translational aspects of viral infection in human brain.

3. MRI for atlasing OPT data: The authors take great care justify each step of their experiment, however it does not seem that they leverage the full potential of the technologies used. For example, atlas registration would allow analysis of OPT data in a much more quantitative way, for example by giving a metric of infection for every atlas-defined region.

The authors completely agree with the reviewer that this is a big advantage of the atlas that comes with the template. However, at the time of submission of this paper, the atlas VOIs coming with the OCUM template were not completely finalized (hence reference to Willekens et al. in preparation in the previous version of the manuscript). Now the complete atlas is finalized (Willekens et al. bioRxiv doi: <https://doi.org/10.1101/2022.11.14.516420>), and we have included atlas-based quantification in the manuscript fig 3a-c,e-l supplementary fig 2 and supplementary table 1.

4. Whole-mount IHC staining with 4 days of incubation time sounds short. Complete, homogenous penetration of labels might not have been achieved. Some of the imaging data indicates this, with stronger signals towards the surface. What steps were taken to ensure uniform penetration of

labels? Is it possible that the OPT tomographic images show heterogenous / superficial labeling?
Could OPT images also be shown in cross section?

Thank you, we now added a supplementary fig 1b of OPT cross section to show the antibody penetration, and we show that we do have staining in deeper regions of the brain.

5. Controls / mock group brain infection OPT data was not noticeably different from infected WT brains as shown in Suppl Fig 1. Moreover, the infected WT data presented in Fig 1 of the main text does not look representative of its cohort. Can the authors explain this?

The reviewer is indeed correct that the WT brain shown in fig 1c, is the most infected as compared to the other four WT brains shown in supplementary fig 1c. In general, we observed large inter-individual differences in both IFNAR and WT mice, this might however be more difficult to assess on MIP images. However, even on the presented MIPs, you can see infection of the entorhinal cortex in two out of four WT brains presented in supplementary fig 1c. Furthermore, we have now performed a quantitative analysis of viral infection using 336 annotated atlas regions (supplementary table 1 and fig 3a) and indeed, found significant infection of the entorhinal cortex in WT mice (mainly driven by the medial entorhinal cortex) as well as, among others, infection of the auditory system. These analyses underline the sensitivity of quantification while visual interpretation can be tricky due to the chosen image representation.

6. The main text does not clearly state how many specimens were used for either experiment. Some information can be found in the captions, but for most experiments the N remains unclear. In some, the N also seems worryingly small and needs discussion. Please clarify.

We thank the reviewer we have now increased the number of cortical slices analyzed for the quantification of infected microglia fig 4e and we have also written all the n in the figure legends.

7. The identification of CD8+ NK cell infiltration and upregulated IFN γ expression in IFNAR $^{-/-}$ but not WT brains is interesting. However, without adoptive transfer or ablation studies the authors cannot conclude whether, in the absence of IFNAR signaling, this response is maladaptive or protective. The molecular mechanisms are weak.

Since the revision we have reanalyzed the snRNA data due to a swapped samples during snRNAseq data analysis. We now show CD8+NK cell infiltration in infected WT mice with FACS analysis and that these cells are the main producers of IFN γ by snRNAseq, we also detect activated microglia with confocal microscopy and upregulation of Ccr1 in WT microglia with snRNAseq after infection. In the Ifnar $^{-/-}$ very little response to infection is induced, microglia are not activated, and they instead become infected.

Minor:

1. Line 92: The authors cite work that is currently 'in preparation'. This makes it challenging to assess the validity and rigor of the OCUM brain template.

We fully agree with the reviewer and we have now completed that manuscript and it is submitted to scientific reports and the full manuscript is also available in BioRxiv that reference is now included in the manuscript.

2. Claim in line 96: “IFN-I response protects meninges and ChP from viral infection”. This conclusion cannot be derived directly from the experiments / data shown. There is an indication that this is true, but causality has not been proven.

We have now changed it to “IFN-I response restricts viral infection in meninges and ChP”.

3. Line 487: Microglia quantification. Analyzing only 6 slices provides hardly sufficient statistical power to formulate a valid statement. Also, how were the slices chosen to be anatomically equivalent such that they can be compared directly?

Thank you for this valid point we have now added more sagittal slices per brain, so that we now have a total of 15 to 21 slices per genotype. These are not consecutive slices but representing different areas of the cerebral cortex.

4. Co-registration of OPT with MRI: The authors claim good alignment - how was this measured?

All performed transformations (coregistration and normalization) were performed according to the Fluorescence-OCUM fusion image creation protocol, described in Willekens et al. (bioRxiv doi: <https://doi.org/10.1101/2022.11.14.516420>). The outcome of and alignment after every applied transformation was reviewed visually and cautiously by two independent readers using the check-registration tool in statistical parametric mapping. This information has been added in the materials and methods.

5. Line 424: “To increase the signal-to-noise ratio for NS5, the pixel intensity range of all images was adjusted to display the minimum and maximum, and a contrast-limited adaptive histogram equalization (CLAHE) algorithm with a tile size of 64 x 64 was applied to projection images acquired in the NS5 channel.”

Contrast enhancement does not increase SNR, so this statement should be corrected. Also, CLAHE boosts contrast locally, but risks enhancing background signal in areas absent of signal. The authors should better explain their reasoning for choosing these post-processing steps and ensure that they are not impacting / altering their conclusions.

The CLAHE script was first described by Hörnblad et al. (ref 74 in manuscript). In this paper, they demonstrate that the application of the CLAHE script increased the relative ratio between objects with weak signal intensity and negative objects without interfering with high positive objects. This implies the possibility of the script to detect weak signals, that would normally be thresholded out during tomographic reconstruction, without enhancing background signal or interfering with objects that exhibit strong signal.

6. Line 436: What are the “phototoxic effects”?

Thank you for pointing this out we have now corrected the sentence to “photobleaching”.

7. Line 542: Inconsistency → euthanasia for Ifnar is stated to have occurred after 3 days, instead of 4-5 days as stated in main text and WT 5 instead of 6-7

We have now clarified in the material and method that two different viral doses were used for the OPT study and the snRNAseq part, the higher dose used for snRNAseq resulted in a faster disease progression and thus earlier humane endpoint.

Reviewers' Comments:

Reviewer #1:

Remarks to the Author:

The authors carefully addressed the reviewers' concerns and recommendations. In particular, they found out that samples in single-nuclei RNA sequencing experiments were swapped. After correction of the genotype of the analyzed samples the data were informative and supported the conclusions drawn. Additional FACS studies of infiltrating cells further substantiated the conclusions drawn from sequencing data. The authors thoroughly worked over the entire manuscript and changed significantly many conclusions. Overall, the manuscript significantly profited from the revision. Now the whole report is conclusive and constitutes a valuable advancement for the field of neuroinfectiology.

Reviewer #2:

Remarks to the Author:

The authors have addressed all of the questions. I appreciate the extra work that they have done to clarify these points.

Reviewer #3:

Remarks to the Author:

The authors did a good job addressing previous concerns and now have a much more cohesive, rigorous study.

One remaining concern is that although the authors identify viral burden regional shifts in WT vs IFNAR1 KO, they focus their later work flow (snRNAseq, flow cytometry, confocal images) purely on the cerebral cortex - a region that is preferentially infected in WT mice. This potentially confounds the lack of inflammatory response and microglia activation they see in IFNAR1 KOs. I think it would be beneficial to at least include confocal images taken from meninges, around ventricular system, and olfactory bulb (all regions more heavily infected in IFNAR1 KOs) to ensure that the lack of inflammatory response in cerebral cortex of mutants is not merely due to regional shifts in infection burden.

Two additional points could be clarified to the reader in the discussion:

Translatability to human: Since the work is motivated to advance the understanding of pathogenesis in humans, it should be discussed more carefully how findings here translate from mice to humans - especially with the controversy around the presence and comparability of the RMS in humans.

Complete antibody penetration in whole mount: The complete penetration of antibodies in the whole mount within 48h still feels very short. The images provided in Suppl. Fig. 1b are helpful, but cannot prove or disprove this concern. It should be mentioned in the methods, what makes the authors confident of a complete and homogeneous antibody penetration that the analysis is based on.

Apart from these minor points, I think this work is a good work fit for Nature Communications.

We thank all the reviewers for their comments, below we address those made by reviewer 3.

Reviewer 3

The authors did a good job addressing previous concerns and now have a much more cohesive, rigorous study.

One remaining concern is that although the authors identify viral burden regional shifts in WT vs IFNAR1 KO, they focus their later work flow (snRNAseq, flow cytometry, confocal images) purely on the cerebral cortex - a region that is preferentially infected in WT mice. This potentially confounds the lack of inflammatory response and microglia activation they see in IFNAR1 KOs. I think it would be beneficial to at least include confocal images taken from meninges, around ventricular system, and olfactory bulb (all regions more heavily infected in IFNAR1 KOs) to ensure that the lack of inflammatory response in cerebral cortex of mutants is not merely due to regional shifts in infection burden.

The cortex is the only region in the brain which is consistently infected in the WT observed by OPT, we also confirm that the cortex has the same infection rate in both WT and Ifnar^{-/-}, this was determined using real time PCR and we also quantified the number of infected cells with confocal microscopy. We also did snRNA sequencing and FACS analysis of the whole cortex and saw no inflammatory response and infiltration of immune cells in IFNAR^{-/-}.

We have now added one additional figure panel in supplemental figure 3, confocal microscopy showing microglial infection in the olfactory bulb, an area with high viral load in Ifnar^{-/-}. This highlights that the tropism toward microglia in Ifnar^{-/-} is observed in other regions of the brain.

Together (snRNAseq, FACS, confocal microscopy, real time PCR, OPT-MRI), this indicates that the lack of inflammatory response in cerebral cortex of Ifnar^{-/-} is not merely due to regional shifts in infection burden.

Two additional points could be clarified to the reader in the discussion:

Translatability to human: Since the work is motivated to advance the understanding of pathogenesis in humans, it should be discussed more carefully how findings here translate from mice to humans - especially with the controversy around the presence and comparability of the RMS in humans.

Thank you for this question, we have now added a sentence in the discussion (line 434-435) to address this concern; "Although our study is not directly translatable to humans, it can serve as a guideline to study translational aspects of viral infection in the human brain."

Complete antibody penetration in whole mount: The complete penetration of antibodies in the whole mount within 48h still feels very short. The images provided in Suppl. Fig. 1b are helpful, but cannot prove or disprove this concern. It should be mentioned in the methods, what makes the authors confident of a complete and homogeneous antibody penetration that the analysis is based on.

We appreciate the question. We used 96 hours incubation for the NS5 antibody, which is in line with studies by others, and saw very good penetration in deep brain areas on tomographic sections of the

3D scanned brains (Supplemental Fig 1b). It is indeed well recognized that the time needed for antibody penetration in whole mount assays may vary due to a variety of factors, including targeted epitopes, tissue type (compactness and make up), size, antibody type, permeabilization protocols etc. Normally, this issue may be investigated by stereological sectioning post 3D scanning, followed by a new antibody label (using the same antibodies) on the sections to avoid the penetration issue, and then match these 2D sections to the tomographic sections. On the other hand, even with this technique it is hard to fully guarantee full antibody penetration since it would, given the factors listed above, require that more or less the entire tissue (brain) is sectioned for each labeling scenario (One of the key reasons for 3D imaging is indeed to omit these tedious and time consuming 2D analyses, with limited possibilities to maintain the 3D context). However, in our case the brains were subjected also to MRI scanning and other applications, making them unsuitable for this purpose. Nevertheless, the apparent staining of deep areas apart, we could as outlined in the manuscript detect essentially the same staining patterns in sections of the same regions, in other brains (not subjected to OPT and MRI processing protocols and scanning) by confocal microscopy. In addition, it should be noted that whole brain OPT obviously does not provide the same resolution as confocal microscopy on sections and that we in such analyses could detect individual cells that were not directly visible in the OPT data. We have in the revised manuscript added a statement about this issue in the methods section reading:

“Of note: the variability of the NS5 signal between viral infections, together with the fact that the tissue was also used for MRI scanning and other applications after OPT acquisition, made it difficult to test antibody penetration efficacy throughout. Still, OPT scanning displayed prominent signal in deep areas of the tissue (see Supplemental Fig 1b), and similar staining patterns were confirmed in separate brains by confocal microscopy”.